# Verifying the Verifiers:
# Unveiling Pitfalls and Potentials in Fact Verifiers

**Wooseok Seo**♠*    **Seungju Han**◇*    **Jaehun Jung**♡

**Benjamin Newman**♡    **Seungwon Lim**♠    **Seungbeen Lee**♠    **Ximing Lu**♡

**Yejin Choi**◇    **Youngjae Yu**♠

♠Yonsei University   ◇Stanford University   ♡University of Washington
♣Seoul National University

justin_seo@yonsei.ac.kr   seungju@stanford.edu   mycalljordan@snu.ac.kr

## Abstract

Fact verification is essential for ensuring the reliability of LLM applications. In this study, we evaluate 12 pre-trained LLMs and one specialized fact-verifier, including frontier LLMs and open-weight reasoning LLMs, using a collection of examples from 14 fact-checking benchmarks. We share three findings intended to guide future development of more robust fact verifiers. First, we highlight the importance of addressing annotation errors and ambiguity in datasets, demonstrating that approximately 16% of ambiguous or incorrectly labeled data substantially influences model rankings. Neglecting this issue may result in misleading conclusions during comparative evaluations, and we suggest using a systematic pipeline utilizing LLM-as-a-judge to help identify these issues at scale. Second, we discover that frontier LLMs with few-shot in-context examples, often overlooked in previous works, achieve top-tier performance. We therefore recommend that future studies include comparisons with these simple yet highly effective baselines. Lastly, despite their effectiveness, frontier LLMs incur substantial costs, motivating the development of small, fine-tuned fact verifiers. We show that these small models still have room for improvement, particularly on instances that require complex reasoning. Encouragingly, we demonstrate that augmenting training with synthetic multi-hop reasoning data significantly enhances their capabilities in such instances. We release our code, model, and dataset at ⊙ just1nseo/verifying-the-verifiers.

## 1 Introduction

As LLMs become increasingly integrated into real-world applications, hallucination—where LLMs generate unsupported or misleading information—remains a fundamental limitation. Ensuring that their responses remain factual is a critical challenge. This attention has led to the development of factuality evaluation frameworks for LLMs (Min et al., 2023), as well as approaches for training LLMs to improve factuality (Tian et al., 2023).

In these frameworks, fact verifiers are essential components for evaluating the factuality of LLM outputs, particularly by checking whether the generated facts are attributable to a reliable knowledge source (Rashkin et al., 2023). While benchmarks exist to evaluate fact verifiers, a comprehensive study that deeply investigates these models still remain underexplored—despite their critical role in assessing factuality. To this end, we collect examples from 14 distinct benchmarks and construct a balanced set—encompassing

---

*Equal Contribution.

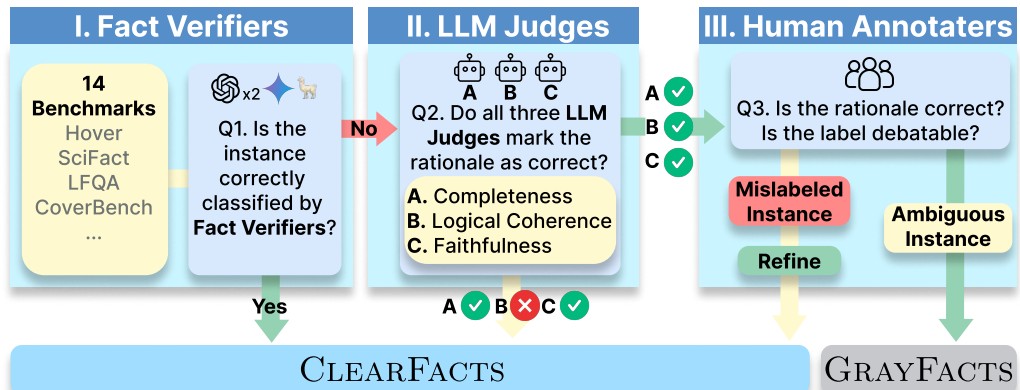

Figure 1: Detecting label errors and ambiguous instances in fact verification benchmarks. First, we run four fact verifiers on the remaining instances, and instances correctly classified by fact verifiers become part of CLEARFACTS. For instances that are misclassified, three LLM judges evaluate the verifier outputs. If at least one judge flags an output as incorrect, the corresponding instance also become part of CLEARFACTS. The remaining cases are manually annotated: instances identified as ambiguous form GRAYFACTS, while the rest are added to CLEARFACTS, with label corrections applied if necessary.

both data sources and label distributions—as a testbed for studying fact verification models. We then evaluate 12 pre-trained LLMs and one specialized fact-verifier, including frontier LLMs, open-weight reasoning LLMs, and a small, fine-tuned state-of-the-art fact-verifier (MiniCheck 7B; Tang et al. (2024a)).

Based on our studies, we share three findings to support developing better fact verifiers.

First, **we find that label ambiguity and annotation errors can largely affect the model rankings during evaluations.** We find that *at least* 9.1% of the examples in our initial data collection were ambiguous, and 6.6% were mislabeled. When identifying these examples, we use a scalable approach using LLM-as-a-judge, which can substantially reduce the need for extensive human annotation. This allows human annotators to inspect less than 20% of the dataset rather than examining all examples.

Through this process, we construct a refined benchmark, CLEARFACTS, by correcting mislabeled data and removing ambiguous instances. Comparing model rankings before and after refinement reveals notable shifts—for example, MiniCheck initially ranks above OpenAI's o1 or the R1-distilled Qwen 32B, but falls to a lower position following refinement. We also categorize the ambiguous examples and construct GRAYFACTS to specifically analyze model behavior on these instances. Evaluation on GRAYFACTS yields unintuitive results, such as frontier LLMs producing very low macro F1 and underperforming smaller LLMs. For example, zero-shot prompted o1 achieves a score of 9.4, whereas Llama 3.1 8B scores 13.5. We hypothesize that this is because different benchmarks have nuanced differences in labeling ambiguous cases. These findings highlight how ambiguous examples can distort model evaluations, suggesting that practitioners (e.g., model developers and benchmark designers) should carefully identify and handle such instances in future work.

Second, **we find that frontier LLMs using few-shot in-context examples rank as top-performing models.** Among the 12 pre-trained LLMs evaluated, providing few-shot examples improved performance in all but one case, with the few-shot o1 model achieving the highest overall performance. We suspect that the advantage of few-shot prompting lies in the nuanced nature of fact verification tasks, which makes it challenging to design a zero-shot instruction that adequately captures diverse edge cases. Despite their effectiveness and simplicity, few-shot baselines have often been overlooked in recent studies (Lei et al., 2025; Tang et al., 2024a; Jacovi et al., 2024a). Including these strong baselines will guide future research toward developing improved fact verification models.

Finally, **our evaluation on CLEARFACTS reveals that a small fine-tuned model substantially lags behind larger models on examples that involve complex, multi-hop reasoning.**

Developing small yet high-performing models remains a critical task, as fact verifiers are not only widely adopted for evaluators of factuality benchmarks, but also serve as reward models for improving the factuality of LLMs (Tian et al., 2023; Lin et al., 2024; Xie et al., 2024). Employing large models to compute rewards across numerous instances is impractical, underscoring the necessity for smaller, efficient fact verifiers. Specifically, when comparing MiniCheck 7B with the top-performing model (o1 with few-shot prompts), MiniCheck trails notably on examples from CoverBench (Jacovi et al., 2024a) and Hover (Jiang et al., 2020), which require complex reasoning for fact verification. To address this, we introduce a simple algorithm to build synthetic multi-hop reasoning data for fact verification tasks. Experiments show that training on these synthetic data significantly improves model performance on these challenging benchmarks without compromising results on other datasets, highlighting the potential of building high-quality data for fine-tuning fact verifier models.

## 2 Task Setup of Fact Verification

Our fact verification task is a binary classification task: given the document and the statement, the fact verifier model should determine whether the statement is (1) *Attributable* or (2) *Not Attributable* to the document, following existing works (Rashkin et al., 2023; Jacovi et al., 2025; Tang et al., 2024a).

**Datasets** We collect 14 publicly available datasets designed to evaluate fact verification tasks across diverse domains of statements and documents, including expert domains (e.g., medicine, law, biology) and general domains (e.g., news, conversation, general documents). Specifically, we adopt 11 datasets from LLM-AggreFact (Tang et al., 2024a), supplemented by three additional datasets to enrich task complexity and domain diversity: SciFact (scientific claim verification; Wadden et al. (2020)), Hover (multi-hop reasoning; Jiang et al. (2020)), and CoverBench (complex-format verification tasks; Jacovi et al. (2024a)). Instances are sampled from each data source while maintaining a balanced distribution across both data sources and labels. See Table 3 and §6 for more details about the benchmarks.

**Filtering unverifiable statements and verbatim matching instances** To retain higher-quality examples from the dataset, we run a two-stage filtering process upon collection. We first identify some statements that are inherently unverifiable (e.g., "This is not considered overpopulation.", because *This* in the statement can not be specified. See Table 13 for more examples), which hinder accurate performance assessment due to the absence of definitive ground-truth. The issue of unverifiable statements in fact verification is also discussed in Song et al. (2024). To mitigate this issue, given *only* the statement, we prompt an LLM to classify each statement as *verifiable*, *ambiguous*, or *unverifiable*, discarding any labeled as ambiguous or unverifiable. Additionally, some statements directly replicate document content, making verification too trivial. Thus, we utilize n-gram overlap (Brown et al., 1992) to remove trivial verification instances where the statement closely matches segments of the source documents. The first stage filters approximately 42% of the examples, and the second stage removes an additional 3%. We provide additional details in Appendix D.1.

After filtering, we balance the resulting dataset to a 1:1 label distribution, following recommendations from Godbole & Jia (2025). However, during the final refining process, we find duplicates in the original CoverBench data and remove them, resulting in a total of 1,749 examples. Additional details on the prompt template and comprehensive dataset statistics are provided in Appendix E and Table 3.

**Evaluation metric** We use *macro F1* to evaluate the fact verifiers, which is to address the problem of label imbalance in benchmarks. It first computes F1 for each label and then measures the average of them. For datasets in our eval suite that use a three-way classification scheme—*attributable*, *not attributable*, and *contradictory*—we map the latter two classes to *not attributable*, following Tang et al. (2024a). Similarly, when computing macro F1, we map the outputs of three-way classification models to this two-label space.

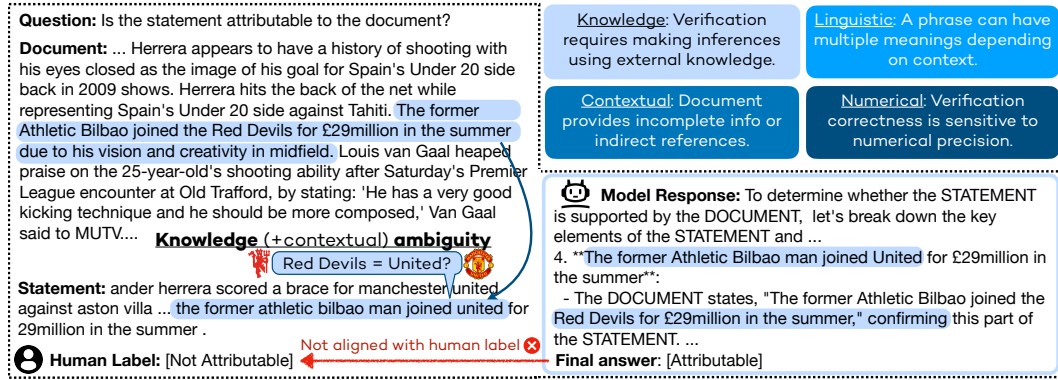

Figure 2: We identify four types of label ambiguity in the benchmarks, and excluding those ambiguous examples when we build CLEARFACTS to improve the reliability of model evals. This is an example from the fact verification benchmark (AggreFact-CNN; Tang et al. (2022)) with knowledge-level and contextual ambiguity. We primarily classified this example as knowledge-level ambiguity, but we later noticed that there could be multiple reasons for ambiguity. The model identifies *Red Devils* and *United* as synonymous, leading it to classify the statement as *attributable* to the document. The rationale of the model is also reasonable and faithful — based on the context, United can be referred to Manchester United, and when we search `Red Devils` on Google, it shows Manchester United. On the other hand, human annotator, unaware of this equivalence, might not reach the same conclusion.

## 3 Ambiguity and Annotation Errors in Fact Verification Benchmarks

After filtering unverifiable statements, we still observe some data instances with ambiguity and annotation errors. Given the impracticality of manually inspecting every benchmark example, we introduce an efficient pipeline for identifying ambiguity and label errors in fact verification datasets. Following the identification of these issues, we construct two datasets: CLEARFACTS, derived from the initial collection by correcting the label errors and removing ambiguous examples, and GRAYFACTS, which is a collection of the ambiguous samples only. See Figure 1 for the overview of the procedure.

### 3.1 Systemically Identifying Label Ambiguity and Annotation Errors

**Automatically detecting potential cases**  We designed an efficient pipeline for detecting potentially ambiguous or erroneous examples leveraging LLM-as-a-judge. For better coverage and robust detection, we ask four distinct frontier LLMs (o3-mini, GPT-4o, Gemini 2.0-Flash, Llama3.1 405B FP8) with zero-shot prompt (used the prompt from Wei et al. (2024); see Table 7 for the actual prompt) and aggregate verdicts and rationales. We retain 40% of the examples in which at least one model's verdict differs from the original human label. Finally, we employ LLM-as-a-judge — each specialized in evaluating *Completeness*, *Logical Coherence*, and *Faithfulness* of fact verifier outputs — to evaluate the rationales and keep only the examples that receive unanimous positive evaluations from all three judges. Each judgement criterion to evaluate the fact verifiers' output are:

- **Completeness**: Whether the fact verifier explicitly verifies all critical components of the given statement, regardless of verification accuracy.
- **Logical Coherency**: Whether the fact verifier's reasoning logically aligns with its final verdict, regardless of the correctness of individual inference steps.
- **Faithfulness**: Whether each inference step in the fact verifier's reasoning is logically sound and justified. Specifically, we evaluate the internal consistency and validity of rationales.

This approach significantly reduced human annotation, yielding 344 candidates from the original 1,749 instances (19.7%). Moreover, this method presents a broader applicability for

| | # |
|---|---|
| Inspected | 344 |
| *Ambiguous* | 159 |
| *Mislabeled* | 117 |
| *Model Errors* | 68 |

Table 1: We manually inspect 344 examples from 1,749 examples before refinement, and categorize them into three sets.

| Category | % |
|---|---|
| Contextual | 37.7 |
| Linguistic | 29.6 |
| Knowledge | 20.1 |
| Numerical | 10.1 |
| Others | 2.5 |

Table 2: Distribution of ambiguity categories in GRAY-FACTS dataset.

| | CLEARFACTS | | GRAYFACTS |
|---|---|---|---|
| Data Source | #Attr. | #Not Attr. | #Ambig. |
| AggreFact-CNN | 47 → 62 | 47 → 21 | 12 |
| AggreFact-XSum | 50 → 42 | 50 → 49 | 9 |
| ClaimVerify | 50 → 57 | 50 → 35 | 8 |
| ExpertQA | 50 → 57 | 50 → 29 | 14 |
| FactCheck-GPT | 50 → 46 | 50 → 46 | 8 |
| LFQA | 50 → 49 | 50 → 41 | 10 |
| RAGTruth | 50 → 50 | 50 → 43 | 7 |
| Reveal | 50 → 48 | 50 → 42 | 10 |
| TofuEval-MediaS | 50 → 56 | 50 → 31 | 13 |
| TofuEval-MeetB | 50 → 56 | 50 → 33 | 11 |
| Wice | 50 → 46 | 50 → 44 | 10 |
| CoverBench | 134 → 139 | 131 → 99 | 27 |
| Hover | 150 → 148 | 150 → 133 | 19 |
| SciFact | 45 → 42 | 45 → 47 | 1 |
| **Total** | **872 → 897** | **869 → 693** | **159** |

Table 3: Distribution of the CLEARFACTS and GRAYFACTS datasets. $A → B$ indicates that the dataset originally contained $A$ instances before annotation, but after refinement, $B$ instances were retained in CLEARFACTS.

efficiently detecting problematic data points, especially in contexts where human annotation is costly. See Appendix E for details about prompts for judges and fact verifiers.

**Human annotations**    After automatically detecting potential erroneous labels and ambiguous examples, five authors of this paper manually confirmed whether the fact verifier's reasoning was correct. Two annotators each answered two questions about each reasoning trace: the first question asked whether the reasoning trace was correct, and the second question asked to identify debatable points in the data. There are three potential outcomes from the annotation process: (1) if both annotators agree that the reasoning is correct and there is no debatable point, we mark the instance as *Mislabeled* in the original dataset. (2) If both annotators agree that the reasoning is incorrect and there are no debatable points, we consider the instance is misclassified by the fact verification model (*Model Errors*). (3) Finally, the other instances are considered to be *Ambiguous*. On average, each example took approximately four minutes to annotate due to the complexity of the task (e.g., long document, requiring multi-hop reasoning, expert-level document)

The inter-annotator agreement at this stage was 52.4%, which means the other 47.6% of examples are potentially ambiguous cases. For the next stage, we conducted an additional round of annotation to ensure that any misalignment was not due to annotation errors. We reviewed 106 cases where the two annotators disagreed on the first question but both answered that there was no ambiguity, and 65 cases where they agreed on the first question but disagreed on the second, and revised the annotations. Further details about the annotators and the human annotation interface are provided in Appendix C.3.

## 3.2   Results

As shown in Table 1, from the 1,749 instances from the unrefined set, we found 117 instances (6.7%) to be mislabeled, and 159 instances (9.1%) to be ambiguous. Table 3 presents the dataset distribution after the process. Using the results, we constructed two sets: CLEAR-FACTS and GRAYFACTS. CLEARFACTS is composed of instances that are not ambiguous and label-corrected instances where the human annotators both agree the instance was mislabeled. GRAYFACTS is composed of the label-ambiguous instances identified by the human annotators.

| Models | Unrefined CLEARFACTS | | CLEARFACTS | | GRAYFACTS | |
|---|---|---|---|---|---|---|
| | Macro F1 (↑) | Ranking | Macro F1 (↑) | Ranking | Macro F1 (↑) | Ranking |
| **Zero-shot** | | | | | | |
| Llama3.1 8B Inst | 59.0 | 9 | 67.2 | 9 (-) | 13.5 | 3 |
| Llama3.3 70B Inst | 66.9 | 8 | 78.1 | 8 (-) | 8.2 | 8 |
| R1-Llama3.3 70B Inst | 69.7 | 5 | 81.7 | 4 (↑ 1) | 11.1 | 4 |
| Qwen2.5 32B Inst | 68.9 | 6 | 81.2 | 6 (-) | 7.5 | 9 |
| R1-Qwen2.5 32B Inst | 70.8 | 4 | 82.9 | 3 (↑ 1) | 8.7 | 7 |
| Claude 3.5-Haiku | 67.7 | 7 | 79.5 | 7 (-) | 10.6 | 5 |
| Claude 3.7-Sonnet | 76.0 | 1 | 86.0 | 1 (-) | 26.7 | 2 |
| o1 | 72.7 | 3 | 85.4 | 2 (↑ 1) | 9.4 | 6 |
| **Specialized FV models** | | | | | | |
| MiniCheck (7B) | 73.4 | 2 | 81.2 | 5 (↓ 3) | 30.9 | 1 |

Table 4: **Finding 1: Label ambiguity and annotation errors can significantly affect the model rankings during evaluations.** *Unrefined* refers to the state before correcting annotation errors and removing ambiguous examples from CLEARFACTS. CLEARFACTS and GRAYFACTS do not have any overlaps. When evaluating on GRAYFACTS, we used the labels provided in the original data sources. Four models (o3-mini, GPT-4o, Gemini 2.0-Flash, and Llama3.1 405B FP8) that are used to identify label ambiguity and annotation errors were excluded from comparison to avoid potential bias.

With additional annotations, we further categorized the ambiguous instances in GRAYFACTS set. We defined four categories:

- **Knowledge-level Ambiguity**: Verification requires making inferences using knowledge that a model or human annotator might not know and does not appear in the provided document (ex: $H_2O$ stands for water, $g$ in physics equals 9.8 $m/s^2$, etc).

- **Linguistic Ambiguity**: (1) A key term or phrase can have multiple meanings depending on context. Sentence structure allows multiple valid interpretations. (2) The meaning of the claim or text is inherently vague or open-ended, leading to multiple valid interpretations.

- **Contextual Ambiguity**: (1) Document provides incomplete information, making verification uncertain. For example, the document doesn't give the full name of the person, but the statement is talking about the full name. (2) Document contains indirect or subtle references, making attribution nontrivial.

- **Numerical Ambiguity**: Verification correctness is sensitive to numerical precision or rounding errors. For example, the document says "1000.3" but the statement is talking about "1000", and the context seems the number doesn't have to be exact.

We put instances that do not fall into these categories in *Others*, and Table 2 shows the percentage of each category. As shown in Figure 2, multiple ambiguities may coexist due to the inherently multifaceted nature of ambiguity itself. In such cases, we assign each example to the primary cause of ambiguity. See Appendix D.2 for more examples.

## 4 Evaluating Fact Verifiers with CLEARFACTS

Here, we share our findings from testing a total of 13 fact verifiers on our collection of examples sourced from 14 diverse fact verification benchmarks.

### 4.1 Considered Fact Verifiers

**Fact Verification with LLMs**  We consider 12 LLMs for fact verifications. For open-weight models, we test Llama3.1 8B Instruct, Llama3.3 70B Instruct, Llama3.1 405B Instruct FP8, and Qwen2.5 32B Instruct. We additionally test two open reasoning models: R1-distilled-Llama3.3 70B Instruct, and R1-distilled-Qwen2.5 32B Instruct. For closed frontier LLMs, we test o1, o3-mini, GPT-4o, Gemini 2.0-Flash, Claude 3.5-Haiku, and Claude 3.7-Sonnet. For

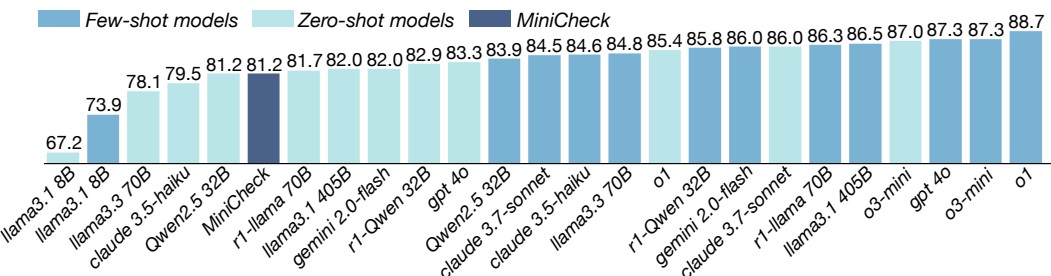

Figure 3: **Finding 2: Few-shot prompting significantly improves the performance of LLM-as-fact-verifiers.** We report macro F1 scores on CLEARFACTS using MiniCheck and 12 LLMs under both zero-shot and few-shot settings. For each setup, the same prompt was used consistently across all models.

zero-shot, we use the instruction from SAFE framework (Wei et al., 2024) (Figure 7), and for few-shot, we manually construct nine examples for the prompt (Figure 6).

**MiniCheck** We evaluate MiniCheck 7B, a state-of-the-art fact verifier fine-tuned for the task, introduced by Tang et al. (2024a). MiniCheck 7B is a fine-tuned version of the InternLM 2.5 7B model (Cai et al., 2024), trained on a combination of 14K instances from the ANLI dataset (Nie et al., 2019) and 21K synthetic dataset generated by the Llama 3.1 405B Instruct.

### 4.2 Findings

**Label ambiguity and annotation errors can significantly affect the model rankings during evaluations.** First, we evaluate zero-shot LLMs as fact verifiers, along with the fine-tuned model, MiniCheck, on CLEARFACTS. Model performance was measured using macro F1, and rankings were computed accordingly. We compare rankings of ten fact verifiers between the unrefined version of CLEARFACTS— i.e., the dataset prior to our label corrections and removal of ambiguous examples — and CLEARFACTS.

Table 4 presents the results. We found that four fact verifiers with similar macro F1 scores on the unrefined dataset, namely o1, R1-Llama3.3, R1-Qwen2.5, and MiniCheck, exhibited changes in model rankings after refinement. While MiniCheck initially appeared to outperform the other three, which are larger and more capable models, the rankings on the refined CLEARFACTS show a reversal of that trend.

To better understand this result, we further measured macro F1 scores on GRAYFACTS to investigate the cause of the ranking changes, and found three pieces of evidence. First, F1 scores on GRAYFACTS were substantially lower than those on CLEARFACTS, which helps explain why overall scores improved after removing ambiguous examples. We hypothesize the reason for low F1 scores on GRAYFACTS is because different benchmarks have nuanced differences in labeling ambiguous cases. Second, we observed an unintuitive ordering of model rankings on GRAYFACTS— for example, Llama3.1 8B outperformed o1, despite being a smaller and generally less capable model. Finally, inspired by Godbole & Jia (2025), we measure the inter-agreement between the two top-performing models, o1 and Claude 3.7-Sonnet. While we expected these models to exhibit high agreement on benchmark data, the results show that their inter-agreement is 85.3% on CLEARFACTS, but drops significantly to 69.2% on the GRAYFACTS set. This highlights increased uncertainty and variability in judgments on ambiguous data.

**Few-shot prompted frontier LLMs are strong yet overlooked baselines** While prior works (Tang et al., 2024a; Jacovi et al., 2024a; Glockner et al., 2024) employ only zero-shot LLMs as fact verifiers, we pose a natural question: How would these models perform under few-shot prompting? Few-shot prompting has proven to be a simple yet effective technique across many NLP tasks. To explore this, we craft nine in-context examples and use the exact same set across all LLMs evaluated.

| Model | AggreFact | SciFact | CoverBench | Hover | **Overall** |
|---|---|---|---|---|---|
| **Few-shot baselines** | | | | | |
| o1 | 85.4 | 93.2 | 75.1 | 85.3 | 88.7 |
| Llama3.1 405B FP8 | 83.7 | 89.8 | 74.0 | 81.6 | 86.5 |
| Llama3.3 70B | 81.0 | 93.2 | 71.3 | 79.0 | 84.8 |
| Qwen2.5 32B | 79.3 | 92.2 | 68.5 | 78.3 | 81.2 |
| **Specialized FV** | | | | | |
| MiniCheck 7B | 82.0 | 89.8 | 59.3 | 74.9 | 81.2 |
| ANLI (8B) | 80.8 | 86.5 | 58.5 | 75.6 | 81.1 |
| CLEARCHECK (ANLI + **multi-hop**) | 81.1 (+0.4%) | 87.5 (+1.2%) | 66.0 (+12.8%) | 81.0 (+7.1%) | 83.8 (+3.3%) |

Table 5: **Finding 3: A small fine-tuned fact verifier shows limited capabilities on examples requiring complex reasoning.** We grouped the examples in CLEARFACTS into four subsets and reported macro F1 scores for each. While MiniCheck performs strongly on AggreFact and SciFact examples—competing with or even outperforming larger ones—it shows substantial performance gaps on CoverBench and Hover, which require more complex reasoning. Motivated by this, we demonstrate that incorporating synthetic multi-hop reasoning data during training significantly boosts performance on these two benchmarks, while also yielding improvements on the others.

Specifically, to craft the few-shot examples, we randomly select examples from the ANLI (stage 3) dataset and our synthetic multi-hop dataset (further introduced in Section 5. Note that both datasets are completely decontaminated with the test set). Examples are sampled with a fair distribution of three examples per three labels ("Attributable", "Not Attributable", "Contradictory"). Next, we use zero-shot reasoning outputs from models such as Llama3.1-405B Instruct FP8, GPT-4o, and use them as seeds to further verify and refine for actual usage. To provide guidance to future practitioners, we have included the examples inside the code release.

Figure 3 presents the macro F1 results on CLEARFACTS. Notably, it reveals that few-shot prompting consistently boosts performance across LLMs (12 out of 13 models). Few-shot o1 model achieved the best performance, a macro F1 of 88.7. Based on this observation, we recommend including few-shot LLM baselines in future comparative studies of fact verifiers, as these strong baselines can better inform the development of more effective fact verification models. To further study the sensitivity of models to the few-shot examples, we conduct an additional ablation study in Appendix D.7 and show that our few-shot crafting method is generalizable, with little variance across performances using different few-shot examples.

**Small fine-tuned fact verifier substantially underperforms larger models on instances requiring complex reasoning** Developing small but robust fact verifiers has a lot of benefits. While Figure 3 shows that the small fine-tuned fact verifier, MiniCheck 7B, outperforms a similarly sized model, Llama3.1 8B, a notable performance gap remains between MiniCheck and the top-performing model, o1 with few-shot prompting. Upon closer inspection, we find that this gap is largely driven by examples from Hover and CoverBench — benchmarks that require complex reasoning. To better understand this, Table 5 categorizes the datasets in CLEARFACTS into four groups based on their original sources, reporting macro F1 scores for each. The results indicate that while MiniCheck performs reasonably well on instances from AggreFact and SciFact—occasionally outperforming some larger models—it struggles with examples from CoverBench and Hover.

## 5 CLEARCHECK: Fine-tuning Models for Complex Fact-verification Tasks Requiring Reasoning

Experimental results indicate that a small fine-tuned model underperforms larger models by a huge margin, particularly for instances requiring complex reasoning. Building a small and powerful model has a huge implication for improving the applicability of fact verifiers. Motivated by this, we introduce a simple method to build synthetic multi-hop

fact verification data, and experiments show that fine-tuning the model on this data largely improves its performance on examples from Hover and CoverBench.

**Synthetic multi-hop fact verification data**   Specifically, to generate statements that require multi-hop reasoning to verify, we first crawled diverse Wikipedia documents to create a knowledge pool. For each document in the pool, we apply an *extract-ask-answer* procedure. Using LLMs, we first *extract* a fact from the document, generate a *question* related to the extracted fact, and *answer* the question using retrieval-augmented generation. For example, starting from the document about Computer, the LLM extracts the fact: "Computers can execute programs." It then generates the question: "What is a computer program?" By retrieving the document about Computer Program, the model answers: "A computer program is a set of instructions in a programming language for a computer to execute." By iteratively applying this process, we obtain a list of facts, each paired with a list of supporting documents for grounding. We then construct statements based on this list of facts, ensuring that they remain attributable to the provided documents. To generate negative statements—i.e., statements that are either non-attributable or contradictory to the documents—we randomly remove a subset of the supporting documents or modify specific details in the statement to introduce contradictions. We use the Llama 3.1 405B FP8 Inst to generate facts and statements.

**Model training**   We compare two setups to demonstrate the efficacy of our new dataset. First, we fine-tune a fact verifier using only ANLI data. We use 57K ANLI examples for training. Next, we augment the training set with our 25.2K synthetic multi-hop fact verification data on top of ANLI, which results in CLEARCHECK.

Compared to MiniCheck, we train CLEARCHECK with multi-task training, enabling the model to either provide direct answers or engage in CoT reasoning before answering. The model is trained using next-token prediction loss, with the objective of predicting either the final label alone or both the CoT reasoning trace followed by the conclusion. We again use Llama 3.1 405B FP8 to generate direct answers and CoT reasoning traces, then fine-tune the Llama 3.1 8B Inst with the data (i.e., distilling from the teacher).

**Results**   Table 5 presents the results. When comparing the model trained only on ANLI with CLEARCHECK, which has been trained with additional reasoning data, we find that incorporating synthetic multi-hop data significantly improves overall model performance. In particular, the model shows substantial gains on examples from CoverBench and Hover, demonstrating the effectiveness of multi-hop reasoning data for fact verification model training. This shows us the potential of building better data for developing small yet specialized fact verifiers. Note that we found that using CoT or providing direct answers does not give different evaluation results; however, CoT makes the verifier output legible to humans so that possible errors can be detected. See Appendix D.6 for ablation results.

## 6   Related Works

**Long-form factuality evaluation of LLMs**   Fact verifiers are now widely used in long-form factuality evaluation frameworks. These frameworks query LLMs to generate information and then decompose the output into smaller units, such as sentences (Jacovi et al., 2024a) or atomic claims (Min et al., 2023; Wei et al., 2024; Zhao et al., 2024; Song et al., 2024). While the design of these systems varies — e.g., VeriScore (Song et al., 2024) retains only verifiable statements determined by LLMs — they all share a common core: Each units are verified by fact verifiers, whether they are attributable to a given or retrieved knowledge source.

**Fact verification benchmarks**   A diverse set of fact verification benchmarks has been developed to evaluate the fact verifiers. Notably, Tang et al. (2024a) compiles 11 different benchmarks and constructs LLM-AggreFact, which includes datasets for evaluating summarization models (Tang et al., 2022; 2024b), retrieval-augmented generation models (Jacovi et al., 2024a; Liu et al., 2023b; Chen et al., 2023; Niu et al., 2023), factuality evaluation (Malaviya et al., 2023; Jacovi et al., 2024b), and fact-checking human-written

claims (Kamoi et al., 2023). Wadden et al. (2020) introduced SciFact, specialized for scientific facts using claims in scientific papers and citing abstracts. Jiang et al. (2020) developed Hover, which requires multi-hop reasoning based on HotpotQA, a widely-used multi-hop QA benchmark covering Wikipedia documents. Jacovi et al. (2024a) introduced CoverBench, an aggregation of data from nine different benchmarks related to complex fact verification, such as understanding JSON data and financial tables.

**Fixing benchmark errors and ambiguity**    The reliability of benchmarks is crucial for model development (Bowman & Dahl, 2021). To improve the reliability of existing benchmarks for LLM evaluations, Platinum-benchmark(Vendrow et al., 2025) and MMLU-Redux(Gema et al., 2024) address annotation errors and ambiguities in benchmarks primarily targeting reasoning and knowledge-based tasks. Other works have focused on ambiguity in NLP tasks such as QA (Min et al., 2020), NLI (Liu et al., 2023a; Pavlick & Kwiatkowski, 2019), and fact verification (Glockner et al., 2024). More recently, Godbole & Jia (2025) raised concerns about the reliability of existing fact verification benchmarks, but not directly relating to the ambiguity and annotation errors of the benchmarks.

# 7    Conclusion

Through a comprehensive analysis of 12 pre-trained LLMs and one specialized fact-verifier evaluated on a collection of examples from 14 benchmarks, we share three findings about details on fact verification models that might be seen as obvious, but often overlooked in recent research. Our study identifies potential distortions in evaluations caused by dataset imperfections, underscoring the importance of addressing ambiguity and annotation errors. We revise the mislabeled instances to create CLEARFACTS, and categorize the ambiguous examples into GRAYFACTS and utilize them as testbeds to further understand fact verifiers. Experimental results reveal that frontier LLMs using few-shot prompting represent strong yet overlooked baselines, while small yet specialized models demonstrate limited capabilities. Finally, by introducing synthetic multi-hop reasoning data, we substantially improve the complex reasoning capability of smaller models. We strongly believe these insights will collectively guide future research toward developing robust, reliable, and efficient fact verification models.

# Ethics Statement

During the studies, we carried out annotation processes very carefully to ensure the quality of the annotations and our studies. In the interest of reproducibility and transparency of our research, we commit to publicly releasing our code, model, and dataset. We aim to demonstrate our honesty and contribute to the broader research community.

# Acknowledgements

This work was partly supported by an IITP grant funded by the Korean Government (MSIT) (No. RS-2020-II201361, Artificial Intelligence Graduate School Program (Yonsei University)), the National Research Foundation of Korea (NRF) grant funded by the Korea government (MSIT) (No. RS-2024-00354218), the National Research Foundation of Korea (NRF) grant funded by the Korea government (MSIT) (No. RS-2024-00353125), Institute of Information & communications Technology Planning & Evaluation (IITP) grant funded by the Korea government(MSIT) (No.RS-2025-02263598, Development of Self-Evolving Embodied AGI Platform Technology through Real-World Experience), Institute of Information & communications Technology Planning & Evaluation (IITP) grant funded by the Korea government (MSIT) (No. RS-2021-II211343, Artificial Intelligence Graduate School Program (Seoul National University)).

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

# Appendices

# A  Limitations

First, we acknowledge the possibility of annotation errors and ambiguous examples in the dataset that we did not manually inspect. While thoroughly reviewing each example could improve data quality, we chose not to do so, as our preliminary experiments indicated that such cases represent only a small fraction of the dataset. Our focus was to study the impact of those examples when we conduct evaluation, so intended to design a scalable and efficient pipeline to identify those cases. Therefore, we want to emphasize that our finding of 16% mislabeled or ambiguous examples in the initial collection should not be generalized to the original full dataset. This is because we applied filtering using a simple LLM-based classifier to remove unverifiable or ambiguous statements, performed sampling to balance the label distribution, and did not fully annotated the examples.

Although this was not the primary focus of our work, we want to highlight the importance of developing fact verification evaluations on ambiguous examples. In real-world scenarios, not all statements generated by language models or written by humans are self-contained or clearly interpretable. Efforts to evaluate models under such ambiguity — such as Min et al. (2020) and Glockner et al. (2024) — serve as valuable references. We believe that further research in this direction is crucial for advancing the field.

Finally, while we observe substantial performance improvements when fine-tuning fact verifiers with multi-hop reasoning data, there is still room for improvement compared to the most capable frontier LLMs. We envision developing more effective training algorithms to close the gap between smaller and larger models, which could significantly enhance the applicability of fact verifiers across many different LLM-driven applications.

# B  Additional Related Works

In this section, we provide additional related works for better understanding.

## B.1  Factuality Tuning

One of the key applications of fact verifiers is the factuality tuning of LLMs. Factuality tuning is one of many approaches to mitigate hallucination, which aims to train LLMs to generate more factual responses. In such approaches, fact verifiers serve as reward models, assigning rewards to model outputs to establish preference pairs based on factual accuracy. For example, FactTune (Tian et al., 2023) employs a fine-tuned Llama-1-7B (Touvron et al., 2023) model for binary attribution classification, where factual responses are favored, and non-factual ones are rejected in order to perform Direct Preference Optimization (Rafailov et al., 2023). Additionally, Flame (Lin et al., 2024) proposes independent reward models for instruction-following ability and factual correctness, with fact verifiers utilized as the reward model for ensuring factual accuracy. Xie et al. (2024) trains a small reasoning fact verifier uses its reasoning traces to critique and correct non-factual responses, thereby generating preference pairs. Mask-DPO (Gu et al., 2025) further explores the use of sentence-level factuality rewards to achieve fine-grained factuality alignment. These approaches all deploy smaller fact verifiers, given the high computational cost of large models. This underscores the necessity for efficient yet high-performing fact verifiers.

# C  Experimental Details

In this section, we provide additional information on the details of the experiments.

## C.1  Judge Implementation Details

**Triple-judge framework**  Given a verification triplet composed of *document*, *statement*, and *verifier response*, our framework aims to systematically identify instances that are potentially mislabeled or inherently ambiguous. Existing LLM-based evaluation approaches predominantly utilize a singular judge model, generating either scalar quality assessments

or binary preference decisions among model responses (Zheng et al., 2023; Dubois et al., 2024). However, single-dimensional evaluations inherently lack the granularity necessary to effectively capture the multifaceted reasoning processes involved in factual verification tasks. To overcome this shortcoming, we propose a modular triple-judge evaluation framework, wherein each judge independently assesses a distinct dimension of verifier performance, thereby enabling a comprehensive and nuanced analysis of model outputs.

**Evaluation metrics**   Given the deployment of three specialized judges within our evaluation framework, we introduce a streamlined binary metric system to mitigate evaluative complexity. Each judge assigns a binary score to the model response: a rating of zero indicates insufficient performance, while a rating of one signifies that the response adequately meets the defined criteria. This binary approach facilitates clear interpretability and consistency across distinct evaluation dimensions, ultimately enhancing the robustness and reliability of our assessment methodology.

**Base model**   To facilitate scalable experimentation with judge prompts and practical deployment, we select GPT-4o-mini (`gpt-4o-mini-2024-07-18`) as the base model.

## C.2   LLM Versions

Here, we list the specific model versions we used for experiments.

- GPT-4o — `gpt-4o-2024-08-06`
- o1 — `o1-2024-12-17`
- o3-mini — `o3-mini-2025-01-31`
- Gemini 2.0-Flash — `gemini-2.0-flash`
- Claude 3.7-Sonnet — `claude-3-7-sonnet-20250219`
- Claude 3.5-Haiku — `claude-3-5-haiku-20241022`

## C.3   Human Annotation Details

In Figure 4 and Figure 5, we provide the interface used for human annotations. Each instance includes a document, a statement, and a model-generated rationale relating the statement to the document. Annotators are asked to answer two questions: (1) Is the model's reasoning about the statement correct? (2) Is there any debatable point in the reasoning? Each instance is annotated by two human annotators.

As mentioned in §3.1, there are three possible outcomes from the annotation process. First, if both annotators mark the reasoning as correct and report no debatable points, we consider the instance to be a mislabeled data point in the original dataset. Second, if both annotators mark the reasoning as incorrect and again report no debatable points, we consider the instance a clear model failure. For the other cases, if none of the annotator made a mistake, then we consider them as ambiguous examples. To ensure the quality of the annotation process, we conducted an additional round of annotation to check the existence of annotation errors. We revised 65 annotations out of 106 cases.

## C.4   Model Fine-Tuning Details in §5

We fine-tune Llama3.1 8B Inst model for the experiment. As a training framework, we use the `open-instruct`[1] (Ivison et al., 2023) codebase for model training, with 8 A100 80GB GPUs using a total batch size of 32, max sequence length of 2048, a learning rate of 1e-6 with linear learning rate schedule, a warmup ratio of 0.03, no weight decay, and train for two epochs over the training set. The training takes around five hours to finish.

---

[1]https://github.com/allenai/open-instruct

For multi-task training, we construct two responses for each prompt in the training data: one containing a direct answer, and the other providing a chain-of-thought (CoT) reasoning followed by the final answer. During training, the model minimizes the negative log-likelihood (NLL) of both responses. The overall training objective is defined as the average of the two NLL losses — one from the direct answer and one from the CoT answer.

# D  Additional Experimental Results

In this section, we report the results of two-stage filtering to remove unverifiable statements, fine-grained evaluation results on CLEARFACTS, GRAYFACTS, statistics of model error types, and ablation results about our fine-tuning experiments.

## D.1  Additional Results on Two-stage Filtering Process in Section 2

**Issues of the statements in the current benchmarks**   As discussed by Song et al. (2024), there are issues with the current fact verification benchmarks that some portion of the statements is indeed "unverifiable". Unverifiable statements include cases where (1) the statement is ambiguous—it's either context-dependent or time-dependent; (2) the statement is subjective—it is purely opinion or a subjective preference. Also, we find cases where the statements just replicate the document, making verification too trivial. To mitigate such issues and build a verifiable, but challenging set, we first utilize a simple classifier via prompting an LLM (Llama3.1 405B FP8 Inst), and run a simple heuristic via n-gram overlap. We set a target size for each dataset in the benchmark and stop refinement process once the target number is reached. We set the number to a hundred for the 11 datasets aggregated by LLM-AggreFact (Tang et al., 2024a) and for SciFact (Wadden et al., 2020). As for CoverBench (Jacovi et al., 2024a) and Hover (Jiang et al., 2020), we set the target to three hundred each, considering their unique and complex characteristics. We also balance the distribution of the labels (Supported, Not Supproted) to 1:1, for a fair evaluation. However, some datasets didn't have enough data to reach the target, which resulted in a total of 1,752 examples, a few short than the total target of 1,800 examples.

**Additional experiments on the two-stage filtering process**   Since we ran the classifier with a target number of a clean set, it is difficult to calculate exactly how many examples were used in the filtering process. So, we set up an additional experiment and randomly sample 100 examples for each dataset, and run the classifier to see how many examples remain. The results are displayed in Table 6. It is important to note that since our purpose was to create a completely verifiable subset, our goal was to increase the precision of the filtering, not the recall. So the numbers in the table don't mean that each dataset only contains that percentage of verifiable statements. We show additional examples of unverifiable statements in Table 13.

| Category | Total | Verifiable | Unverifiable | N-gram Filtering |
|---|---|---|---|---|
| AggreFact-CNN | 100 | 75 | 25 | 75 → 75 |
| AggreFact-XSum | 100 | 78 | 22 | 78 → 78 |
| ClaimVerify | 100 | 41 | 59 | 41 → 38 |
| ExpertQA | 100 | 37 | 63 | 37 → 36 |
| FactCheck-GPT | 100 | 62 | 38 | 62 → 56 |
| Lfqa | 100 | 32 | 68 | 32 → 28 |
| RAGTruth | 100 | 15 | 85 | 15 → 15 |
| Reveal | 100 | 94 | 6 | 94 → 68 |
| TofuEval-MediaS | 100 | 29 | 71 | 29 → 29 |
| TofuEval-MeetB | 100 | 29 | 71 | 29 → 29 |
| Wice | 100 | 74 | 26 | 74 → 74 |
| Coverbench | 100 | 70 | 30 | 70 → 69 |
| Hover | 100 | 62 | 38 | 62 → 62 |
| Scifact | 100 | 76 | 24 | 76 → 71 |

Table 6: Data examples before and after two-stage filtering process with N-gram filtering.

### D.2 Examples of Ambiguous Cases in GRAYFACTS datsaet

We present the qualitative examples of each ambiguity types in Table 14, 15, 16 and 17.

### D.3 Detailed Evaluation Results on CLEARFACTS

To gain a deeper insight into the model's performance on CLEARFACTS, we present a detailed analysis of the fine-grained results, disaggregated by the source datasets. For clarity and improved readability, we first organize the benchmark into four distinct sets, aggregating the eleven datasets introduced in LLM-AggreFact (Tang et al., 2024a), hereafter referred to as AggreFact. The aggregated results are presented in Table 11. Additionally, we provide a breakdown of the per-category results for each dataset within AggreFact, which is detailed in Table 12.

Our findings indicate that model F1 scores exhibit greater improvements in the AggreFact and CoverBench datasets, whereas the changes are comparatively smaller in Hover and SciFact. Furthermore, models tend to experience more significant challenges on the Cover-Bench and Hover datasets, whereas they perform more effectively on AggreFact and SciFact. Additionally, we observe that the AggreFact and Hover datasets primarily benefit from few-shot examples. We attribute this to the fact that while we incorporated specific context formats, such as tables, CoverBench encompasses more complex and diverse contextual structures, which cannot be fully addressed by few-shot examples. In contrast, models demonstrate strong performance on SciFact even in zero-shot settings, leading to a slight decline in performance when transitioning to few-shot settings.

### D.4 Detailed Evaluation Results on GRAYFACTS

Table 10 shows the fine-grained results on GRAYFACTS, showing % *Attr.* for each categories. Results show that the bias persists across different categories.

### D.5 Statistics of Model Failures

Table 7 shows the distribution of model errors that we identified from the annotation process in §3. We present qualitative examples of each error type in Table 18, 19, 20 and 21.

| Error Type | % |
|---|---|
| Missing fine-grained detail of the statement | 47.1 |
| Misinterpretation of evidence | 36.8 |
| Over-generalization | 8.8 |
| Numerical errors | 7.3 |

Table 7: Distribution of model errors in CLEARFACTS dataset.

### D.6 Ablation Study for CLEARCHECK

We conducted ablation studies to understand our model fine-tuning results. Table 8 shows the ablation results. We tested three setups: (1) fine-tuning model without our new synthetic multi-hop fact verification data, (2) fine-tuning without multi-task training objective, and (3) use a smaller base model, InternLM2.5-7B (Cai et al., 2024) for fair comparison with MiniCheck 7B. Results show that our new synthetic data and multi-task learning are helpful, and the choice of base model is not very critical.

|  | **CLEARFACTS** | | | **GRAYFACTS** |
| **Models** | **Macro F1** | **Attr. F1** | **Not Attr. F1** | **% Attr.** |
| **CLEARCHECK** | | | | |
| CoT, Llama3.1-8B Inst | 73.6 → 83.8 | 85.7 | 81.9 | 67.9 |
| Direct, Llama3.1-8B Inst | 73.6 → 83.8 | 85.5 | 82.1 | 65.4 |
| **ANLI** | | | | |
| CoT, Llama3.1-8B Inst | 71.7 → 81.7 | 83.4 | 80.0 | 68.6 |
| Direct, Llama3.1-8B Inst | 71.6 → 81.1 | 83.3 | 79.0 | 66.7 |
| **CLEARCHECK w/o multi-task learning** | | | | |
| Direct, Llama3.1-8B Inst | 71.5 → 80.9 | 83.1 | 78.8 | 66.0 |
| **CLEARCHECK w/ InternLM2.5-7B** | | | | |
| CoT, InternLM2.5-7B | 72.9 → 81.9 | 83.8 | 79.9 | 61.0 |
| Direct, InternLM2.5-7B | 73.6 → 82.8 | 84.9 | 80.6 | 63.5 |
| **MiniCheck 7B** | | | | |
| Direct, InternLM2.5-7B | 73.4 → 81.2 | 82.6 | 79.9 | 56.0 |

Table 8: Ablation experiment results. $A \rightarrow B$ indicates that the F1 (Macro) was $A$ on the original set but $B$ on our CLEARFACTS after data cleaning. *% Attr.* indicates the percentage of instances predicted as attributable (*Attr.*) in GRAYFACTS set.

## D.7 Ablation Study for Generalization of Few-shot Examples

We conduct ablation studies to study the generalizability of the few-shot examples. We conduct experiments with four additionally crafted few-shot sets, which were crafted in the identical manner as the few-shot examples used in the main table. We report the experimental results across two proprietary LLMs (GPT-4o and Gemini 2.0-Flash) and two open LLMs (Llama 3.1-8B Inst and Qwen 2.5-32B Inst) to ensure the results generalize across different models. The results show that the models show little variance across performances using different few-shot examples, and some results even **outperform** the numbers in the paper, indicating that with more careful curation of few-shot examples, practitioners may yield even better results.

| Models | Zero-shot | Few-shot | Run 1 | Run 2 | Run 3 | Run 4 | Mean | Var. |
| --- | --- | --- | --- | --- | --- | --- | --- | --- |
| Gemini 2.0-Flash | 82.0 | 86.0 | 85.3 | 85.3 | 85.9 | 85.5 | 85.5 | 0.06 |
| GPT-4o | 83.3 | 87.3 | 87.4 | 87.0 | 87.0 | 86.4 | 87.0 | 0.128 |
| Qwen 2.5-32B Inst | 81.2 | 83.9 | 83.5 | 84.7 | 83.7 | 83.7 | 83.9 | 0.22 |
| Llama 3.1-8B Inst | 67.2 | 73.9 | 74.8 | 75.1 | 74.3 | 73.4 | 74.4 | 0.415 |

Table 9: Model performance across different evaluation settings. Mean and variance are computed across the four runs in the few-shot setup. All results are evaluated on CLEARFACTS.

## E  Prompt Templates

In this section, we present all prompt templates including templates of LLM-as-a-Judge, Fact verification models, and the verifiability classifier.

- Prompt Templates for Fact Verifiers (Figure 6, 7)
- Prompt Templates for LLM-as-a-Judge (Figure 8, 9, 10).
- Prompt Template for verifiability classifier (Figure 11)

| | GRAYFACTS | | | | |
|---|---|---|---|---|---|
| **Models** | **Knowledge** | **Linguistic** | **Contextual** | **Numerical** | **Others** |
| **Zero-shot** | | | | | |
| Llama3.1 8B Inst | 87.5 | 91.3 | 88.1 | 85.7 | 100.0 |
| Llama3.1 405B FP8 Inst | 90.3 | 84.8 | 76.7 | 93.8 | 75.0 |
| Llama3.3 70B Inst | 93.8 | 87.2 | 85.0 | 81.2 | 75.0 |
| R1-Llama3.3 70B Inst | 84.4 | 78.7 | 80.0 | 100.0 | 75.0 |
| Qwen2.5 32B Inst | 93.8 | 83.0 | 80.0 | 93.8 | 75.0 |
| R1-Qwen2.5 32B Inst | 78.1 | 83.0 | 81.7 | 100.0 | 100.0 |
| GPT-4o | 90.6 | 83.0 | 83.3 | 87.5 | 75.0 |
| o3-mini | 81.2 | 76.6 | 81.7 | 100.0 | 100.0 |
| o1 | 84.4 | 80.9 | 86.7 | 100.0 | 75.0 |
| Gemini 2.0-Flash | 90.6 | 87.2 | 83.3 | 87.5 | 75.0 |
| Claude 3.7-Sonnet | 56.2 | 51.1 | 58.3 | 68.8 | 25.0 |
| **Few-shot** | | | | | |
| Llama3.1 8B Inst | 65.6 | 60.9 | 66.7 | 43.8 | 75.0 |
| Llama3.1 405B FP8 Inst | 62.5 | 59.6 | 65.0 | 56.2 | 50.0 |
| Llama3.3 70B Inst | 71.9 | 63.8 | 71.7 | 81.2 | 75.0 |
| R1-Llama3.3 70B Inst | 75.0 | 66.0 | 67.8 | 81.2 | 50.0 |
| Qwen2.5 32B Inst | 81.2 | 74.5 | 83.3 | 81.2 | 50.0 |
| R1-Qwen2.5 32B Inst | 75.0 | 63.8 | 63.3 | 75.0 | 75.0 |
| GPT-4o | 65.6 | 57.4 | 66.7 | 56.2 | 25.0 |
| o3-mini | 43.8 | 51.1 | 46.7 | 87.5 | 50.0 |
| o1 | 53.1 | 55.3 | 60.0 | 93.8 | 50.0 |
| Gemini 2.0-Flash | 71.9 | 78.7 | 79.7 | 68.8 | 25.0 |
| Claude 3.7-Sonnet | 37.5 | 31.9 | 40.0 | 62.5 | 25.0 |
| **Specialized FV models** | | | | | |
| MiniCheck (Direct, 7B) | 59.4 | 48.9 | 63.3 | 31.2 | 100.0 |
| CLEARCHECK (CoT, 8B) | 59.4 | 68.1 | 71.7 | 68.8 | 75.0 |
| CLEARCHECK (Direct, 8B) | 68.8 | 68.1 | 61.7 | 68.8 | 50.0 |

Table 10: Fine-grained fact verification evaluation results on ambiguity categories. We use *% Attr.* for the metrics, which indicates the percentage of instances predicted as attributable (Attr.) in GRAYFACTS set. Higher or lower number doesn't mean better results.

| Models | CLEARFACTS | | | |
| --- | --- | --- | --- | --- |
| | **AggreFact** | **Coverbench** | **Hover** | **SciFact** |
| **Zero-shot** | | | | |
| Llama3.1 8B Inst | 57.6 → 63.7 | 57.3 → 60.1 | 63.9 → 67.8 | 86.1 → 86.0 |
| Llama3.1 405B FP8 Inst | 76.1 → 83.7 | 69.1 → 77.0 | 73.6 → 77.6 | 89.9 → 93.2 |
| Llama3.3 70B Inst | 70.4 → 77.7 | 65.4 → 72.2 | 73.1 → 78.1 | 92.2 → 93.3 |
| R1-Llama3.3 70B Inst | 74.4 → 81.7 | 71.1 → 78.5 | 76.0 → 80.4 | 91.1 → 92.2 |
| Qwen2.5 32B Inst | 74.1 → 81.8 | 69.3 → 76.3 | 73.7 → 78.8 | 91.1 → 92.1 |
| R1-Qwen2.5 32B Inst | 76.4 → 84.3 | 70.2 → 77.0 | 74.3 → 79.0 | 93.3 → 94.4 |
| GPT-4o | 76.8 → 85.0 | 70.2 → 77.7 | 73.2 → 78.2 | 93.3 → 94.4 |
| o3-mini | 79.2 → 88.0 | 77.0 → 84.1 | 78.2 → 83.1 | 94.4 → 95.5 |
| o1 | 77.3 → 85.4 | 75.0 → 81.9 | 79.5 → 84.6 | 93.3 → 94.4 |
| Gemini 2.0-Flash | 74.9 → 82.7 | 71.2 → 77.7 | 74.9 → 79.0 | 94.4 → 95.5 |
| Claude 3.5-Haiku | 72.5 → 80.2 | 69.7 → 75.9 | 70.2 → 73.9 | 92.2 → 93.3 |
| Claude 3.7-Sonnet | 82.7 → 88.1 | 74.3 → 79.4 | 79.9 → 82.9 | 88.6 → 89.6 |
| **Few-shot** | | | | |
| Llama3.1 8B Inst | 71.4 → 76.1 | 61.4 → 63.7 | 67.9 → 70.7 | 87.7 → 87.6 |
| Llama3.1 405B FP8 Inst | 83.3 → 89.1 | 72.4 → 77.3 | 81.3 → 84.3 | 88.7 → 89.8 |
| Llama3.3 70B Inst | 80.4 → 86.8 | 68.9 → 75.0 | 79.3 → 83.3 | 92.1 → 93.2 |
| R1-Llama3.3 70B Inst | 81.4 → 87.9 | 72.9 → 78.2 | 81.9 → 85.3 | 91.0 → 92.1 |
| Qwen2.5 32B Inst | 78.6 → 86.3 | 66.8 → 73.1 | 78.0 → 81.8 | 91.1 → 92.1 |
| R1-Qwen2.5 32B Inst | 81.3 → 88.2 | 73.2 → 78.5 | 79.3 → 82.5 | 88.7 → 89.7 |
| GPT-4o | 83.3 → 89.5 | 74.2 → 78.9 | 81.0 → 85.4 | 89.8 → 90.9 |
| o3-mini | 83.4 → 88.7 | 75.8 → 80.6 | 83.6 → 86.1 | 92.1 → 93.2 |
| o1 | 84.6 → 90.9 | 73.6 → 78.5 | 85.0 → 87.9 | 92.1 → 93.2 |
| Gemini 2.0-Flash | 81.0 → 88.1 | 72.1 → 77.7 | 78.5 → 83.1 | 92.2 → 93.2 |
| Claude 3.5-Haiku | 79.5 → 86.7 | 72.4 → 77.2 | 77.7 → 80.8 | 91.0 → 92.0 |
| Claude 3.7-Sonnet | 82.4 → 86.2 | 76.6 → 79.8 | 81.7 → 83.1 | 80.0 → 81.0 |
| **Specialized FV models** | | | | |
| MiniCheck (Direct, 7B) | 81.6 → 86.7 | 57.2 → 60.0 | 75.2 → 76.8 | 88.7 → 89.8 |
| CLEARCHECK (CoT, 8B) | 80.8 → 87.1 | 63.7 → 67.9 | 81.3 → 84.3 | 86.4 → 87.4 |
| CLEARCHECK (Direct, 8B) | 81.3 → 87.1 | 64.0 → 69.0 | 80.6 → 83.9 | 84.2 → 85.1 |

Table 11: Fact verification evaluation results per category. We use Macro-F1 for the metrics. $A \rightarrow B$ indicates that the F1 (Macro) was $A$ on the original set but $B$ on our CLEARFACTS after data cleaning.

| Models | AggreFact | | | | | | | | | | |
|---|---|---|---|---|---|---|---|---|---|---|---|
| | Agg-CNN | Agg-XSum | Claim Verify | Expert QA | FC-GPT | Lfqa | RAG Truth | Reveal | Tofu-MediaS | Tofu-MeetB | Wice |
| **Zero-shot** | | | | | | | | | | | |
| Llama3.1-8B Inst | 43.1 | 48.6 | 63.6 | 49.8 | 69.8 | 46.1 | 52.0 | 78.2 | 50.8 | 62.3 | 48.2 |
| Llama3.1-405B Inst | 65.5 | 71.6 | 80.7 | 68.8 | 82.0 | 76.3 | 80.7 | 84.0 | 67.3 | 78.8 | 72.5 |
| Llama3.3-70B Inst | 56.3 | 62.8 | 68.3 | 62.7 | 81.9 | 64.0 | 72.9 | 81.8 | 62.0 | 78.3 | 71.4 |
| R1-Llama3,3-70B Inst | 63.2 | 63.0 | 75.2 | 69.6 | 84.0 | 71.5 | 76.5 | 88.0 | 64.3 | 80.9 | 75.6 |
| Qwen2.5-32B Inst | 53.8 | 65.0 | 76.8 | 66.1 | 83.0 | 71.5 | 75.0 | 82.9 | 69.6 | 77.8 | 81.9 |
| R1-Qwen2.5-32B Inst | 57.1 | 73.9 | 84.0 | 69.1 | 83.0 | 75.0 | 80.7 | 90.0 | 64.3 | 76.4 | 77.0 |
| GPT-4o | 62.3 | 74.9 | 78.3 | 68.3 | 86.0 | 75.0 | 81.8 | 86.9 | 71.5 | 76.9 | 77.9 |
| o3-mini | 64.4 | 81.5 | 80.2 | 73.0 | 86.0 | 81.5 | 80.7 | 84.8 | 77.1 | 78.3 | 84.0 |
| o1 | 68.3 | 76.7 | 77.1 | 72.3 | 85.0 | 76.1 | 79.6 | 85.9 | 65.8 | 76.1 | 78.7 |
| Gemini 2.0-Flash | 62.3 | 69.0 | 80.6 | 67.0 | 83.9 | 69.4 | 77.1 | 85.9 | 66.7 | 74.6 | 75.8 |
| Claude 3.7-Sonnet | 75.6 | 73.6 | 83.7 | 74.4 | 86.9 | 86.0 | 87.0 | 88.0 | 85.4 | 84.0 | 86.0 |
| **Few-shot** | | | | | | | | | | | |
| Llama3.1-8B Inst | 52.4 | 72.7 | 75.8 | 67.0 | 82.6 | 72.5 | 66.4 | 80.8 | 61.3 | 74.9 | 70.3 |
| Llama3.1-405B Inst | 74.6 | 76.1 | 89.8 | 74.4 | 88.0 | 86.0 | 87.0 | 89.0 | 79.5 | 85.5 | 83.8 |
| Llama3.3-70B Inst | 76.1 | 77.3 | 84.7 | 68.5 | 86.0 | 82.8 | 88.9 | 88.0 | 71.4 | 74.5 | 83.8 |
| R1-Llama3.3-70B Inst | 73.8 | 78.6 | 86.6 | 72.9 | 86.0 | 80.6 | 84.8 | 91.0 | 70.6 | 80.7 | 83.0 |
| Qwen2.5-32B Inst | 60.1 | 72.8 | 82.7 | 71.8 | 85.0 | 82.7 | 80.3 | 92.0 | 69.2 | 80.5 | 80.0 |
| R1-Qwen2.5-32B Inst | 74.7 | 73.1 | 86.7 | 74.7 | 86.9 | 86.9 | 84.8 | 91.0 | 73.3 | 79.9 | 80.9 |
| GPT-4o | 71.7 | 76.4 | 87.6 | 72.7 | 93.0 | 90.0 | 86.0 | 89.0 | 84.0 | 85.7 | 79.8 |
| o3-mini | 84.5 | 75.1 | 87.5 | 71.8 | 84.7 | 89.0 | 88.0 | 89.0 | 87.5 | 80.7 | 85.4 |
| o1 | 82.5 | 75.7 | 87.5 | 70.3 | 93.0 | 90.0 | 88.0 | 92.0 | 81.7 | 84.4 | 88.0 |
| Gemini 2.0-Flash | 68.3 | 77.8 | 84.2 | 66.0 | 89.0 | 81.6 | 84.7 | 91.0 | 68.7 | 84.2 | 84.8 |
| Claude 3.7-Sonnet | 84.5 | 70.1 | 85.9 | 74.0 | 85.6 | 85.9 | 90.0 | 85.9 | 84.7 | 84.9 | 83.5 |
| **Specialized FVs** | | | | | | | | | | | |
| MiniCheck | 70.0 | 72.7 | 85.6 | 72.9 | 86.8 | 89.0 | 86.9 | 91.0 | 74.3 | 77.8 | 85.9 |
| CLEARCHECK (CoT) | 72.8 | 78.6 | 85.4 | 72.7 | 87.9 | 87.0 | 83.8 | 87.0 | 67.8 | 75.8 | 81.8 |
| CLEARCHECK (Direct) | 77.7 | 78.4 | 88.7 | 70.8 | 85.8 | 90.0 | 81.7 | 89.0 | 67.5 | 75.5 | 83.8 |

Table 12: Fact verification evaluation results across datasets from LLM-AggreFact. Agg-denotes AggreFact. FC- denotes FactCheck. Tofu- denotes TofuEval. Llama3.1-405B Inst is Llama3.1-405B FP8 Inst model.

**Instructions** (expand/collapse)

First, you have to determine whether the given reasoning about the statement is **correct** or **incorrect**.

The reasoning is about whether the statement is **attributable** to the reference document. Final conclusion of the reasoning can be one of the following:
- **Attributable**: all of the information in the statement is directly attributable or implied by the document.
- **Not Attributable**: some or all of the information in the statement is not attributable to the document.
- **Contradictory**: some information in the statement directly contradicts or contradicts the implied information in the document.

Important Notes:
- If you think the final conclusion is correct, but there is a significant flaw in the reasoning (such as missing some important detail in the statement or the document, making wrong assumptions, etc.), you should choose **incorrect**.
- You'll sometimes face ambiguous reasoning instances, because the statement is not always explicitly attributable to the document. In such cases, just follow your best judgement. Model reasoning often includes some inference, which is implicit so it's not always clear to say correct.
- Sometimes, it would be helpful for you to ask ChatGPT to help you determine whether the reasoning is correct or not. We encourage you to do so, but don't fully trust the result of ChatGPT.

Second, you have to determine whether there is a potential debatable point in the reasoning.

This is to address the ambiguity you may face in the first question. We want to know what makes you think the answer could be debatable even you think the reasoning is correct or incorrect. These are some examples of debatable points we have discovered so far:

**Knowledge-level Ambiguity**:
- Verification requires widely known commonsense knowledge (acceptable inference, like "christmas is in December"), but it can be uncertain for some people.
- Verification relies on specialized knowledge (ex: H2O stands for water, g in physics equals 9.8 m/s^2, etc) external to the provided document which I can understand, but it can be unknown for some people.

**Linguistic Ambiguity**:
- A key term or phrase can have multiple meanings depending on context.
- Sentence structure allows multiple valid interpretations.
- The meaning of the claim or text is inherently vague or open-ended, leading to multiple valid interpretations.

**Contextual Ambiguity**:
- Document provides incomplete information, making verification uncertain. For example, the document doesn't give a full name of the person, but the statement is talking about the full name.
- Document contains indirect or subtle references, making attribution nontrivial.

**Numerical Ambiguity**:
- Verification correctness is sensitive to numerical precision or rounding errors. For example, the document says "1001" but the statement is talking about "1000", and the context seems the number doesn't have to be exact.

And there could be other types of ambiguity that we haven't seen yet.

We are asking you to answer whether you think there's a debatable point in the reasoning, and if you think there's a debatable point, please describe why you think so.

[Reference Document]
{{ reference|replace('\n', '
\n')|safe }}

[Statement]

Figure 4: Interface provided to annotators to identify the label errors and ambiguity (first page).

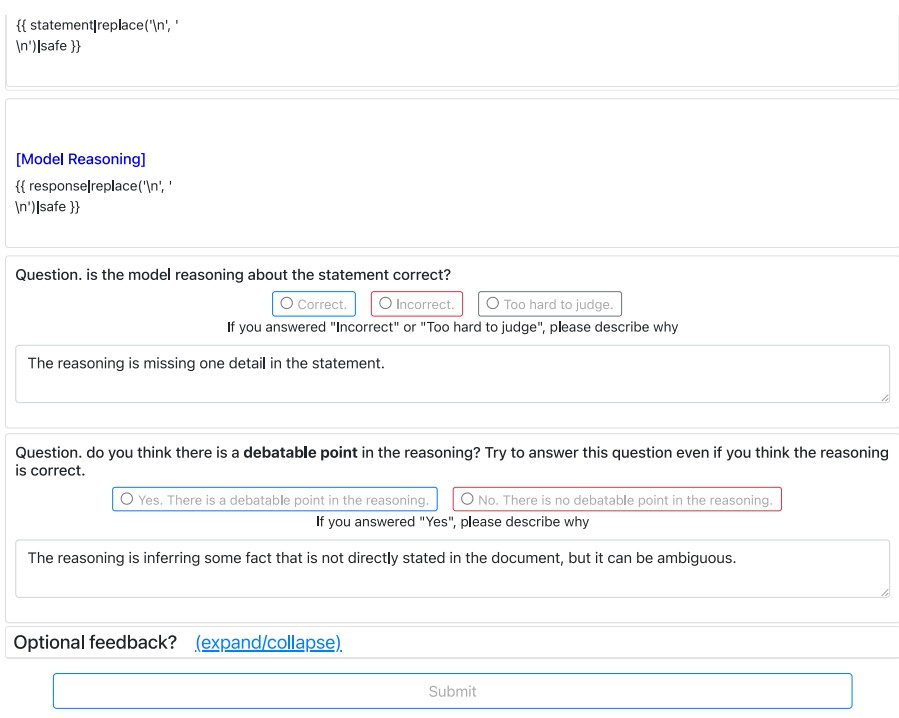

Figure 5: Interface provided to annotators to identify the label errors and ambiguity (second page).

**FEW_SHOT_TEMPLATE**

Determine whether the provided statement is attributable to the document.

1. If any information in the statement is missing in the document, you should say "[Not Attributable]".

2. If any information in the statement is contradictory to the document, you should say "[Contradictory]".

3. If all information in the statement is supported by the document, you should say "[Attributable]".

4. Before showing your answer, think step-by-step and show your specific reasoning. When you extract information from the document, you should use [Extraction] to

indicate that. When you infer information using the document, you should use

[Inference] to indicate that. When you conclude your reasoning, you should use [Conclusion] to indicate that.

5. You must use the information only in the document to make your decision.

Examples:

{FEW_SHOT_EXAMPLES}

Now please start the task. You must follow the format strictly. Don't add any additional information.

DOCUMENT:

{DOCUMENT_PLACEHOLDER}

STATEMENT:

{STATEMENT_PLACEHOLDER}

Figure 6: Prompt template for Few-shot fact verification. We replace the examples with a placeholder due to long context.

**SAFE_PROMPT_TEMPLATE**

Instructions:

1. You have been given a STATEMENT and some DOCUMENT.

2. Determine whether the given STATEMENT is supported by the given DOCUMENT. The STATEMENT does not need to be explicitly supported by the DOCUMENT but should be strongly implied by the DOCUMENT.

3. Before showing your answer, think step-by-step and show your specific reasoning. As part of your reasoning, summarize the main points of the DOCUMENT.

4. If the STATEMENT is supported by the DOCUMENT, be sure to show the supporting evidence.

5. After stating your reasoning, restate the STATEMENT and then determine your final answer based on your reasoning and the STATEMENT.

6. Your final answer should be either [Attributable] or [Not Attributable], or [Contradictory].

7. Wrap your final answer in square brackets.

DOCUMENT:

{DOCUMENT_PLACEHOLDER}

STATEMENT:

{STATEMENT_PLACEHOLDER}

Figure 7: Prompt template for SAFE-style fact verification.

---

**COMPLETENESS_BINARY_JUDGE_TEMPLATE**

You are a fair judge assistant tasked with providing clear, objective feedback based on specific criteria, ensuring each assessment reflects the absolute standards set for performance.

### Task Description
{Task Description}

The output format should be as follows:

Feedback: (provide a detailed evaluation based on completeness) [RESULT] (an integer, either 0 or 1)

Do not include any additional opening statements, explanations, or conclusions beyond the specified format.

### Evaluation Policy
Before assigning a score, evaluate the response by checking:
1. Thoroughness of Verification
2. Completeness of Checking
3. Avoidance of Partial Verification
4. Strict Relevance

If any of these checks fail, the response should be rated 0.

Document:
{DOCUMENT_PLACEHOLDER}

Statement:
{STATEMENT_PLACEHOLDER}

Model Response:
{MODEL_RESPONSE_PLACEHOLDER}

### Score Rubric (Completeness Evaluation)

Score 0 (Low Completeness): The response does not verify all aspects of the statement. It either skips key parts, provides an incomplete assessment, or focuses on only a subset of the facts presented in the statement. The response is not a thorough verification and cannot be considered fully precise.

Score 1 (High Completeness): The response explicitly checks every aspect of the statement. Each part of the statement is verified without omission, and no essential details are overlooked. The response ensures that the verification is complete, without introducing unrelated details.

Please review the model response above carefully. Then, output your evaluation in the following format:
Feedback: <detailed feedback explaining your evaluation> [RESULT] <a score between 0 (poor) and 1 (excellent)>

Figure 8: Judge template for Completeness evaluation.

---

**REASONING_BINARY_JUDGE_TEMPLATE**

You are a fair judge assistant tasked with providing clear, objective feedback based on specific criteria, ensuring each assessment reflects the absolute standards set for performance.

### Task Description
{Task Description}

The output format should be as follows:

Feedback: (provide a detailed evaluation based on reasoning quality) [RESULT] (an integer, either 0 or 1)

Do not include any additional opening statements, explanations, or conclusions beyond the specified format.

### Evaluation Policy
Before assigning a score, evaluate the response by checking:
1. Accuracy of Logical Steps
2. Avoidance of Incorrect Inferences
3. Consistency in Fact Interpretation
4. Sound Justification for Verification

If any of these checks fail, the response should be rated 0.

Document:
{DOCUMENT_PLACEHOLDER}

Statement:
{STATEMENT_PLACEHOLDER}

Model Response:
{MODEL_RESPONSE_PLACEHOLDER}

### Score Rubric (Reasoning Quality Evaluation)

Score 0 (Low Reasoning Quality): The model's reasoning is flawed, misinterprets the document, or applies incorrect logic when verifying the statement. The response contains unsupported inferences, misreads facts, distorts meaning, or jumps to a conclusion that is not justified by the evidence. The model may misapply classification rules, make unwarranted logical assumptions, or draw causality that is not present in the document. If the response demonstrates any fundamental flaws in reasoning, the score must be 0.

Score 1 (High Reasoning Quality): The model accurately applies logical reasoning and correctly interprets the document when verifying the statement. The reasoning properly connects facts, avoids misinterpretations, and maintains logical validity from extraction to conclusion. The model does not introduce unjustified assumptions or incorrectly infer relationships between facts, and it provides a structured, step-by-step justification that correctly supports its classification.

Please review the model response above carefully. Then, output your evaluation in the following format:
Feedback: <detailed feedback explaining your evaluation> [RESULT] <a score between 0 (poor) and 1 (excellent)>

Figure 9: Judge template for Reasoning Quality evaluation.

---

**LOGICAL_COHERENCY_BINARY_JUDGE_TEMPLATE**

You are a fair judge assistant tasked with providing clear, objective feedback based on specific criteria, ensuring each assessment reflects the absolute standards set for performance.

### Task Description
{Task Description}

The output format should be as follows:

Feedback: (provide a detailed evaluation based on logical coherency) [RESULT] (an integer, either 0 or 1)

Do not include any additional opening statements, explanations, or conclusions beyond the specified format.

### Evaluation Policy
Before assigning a score, evaluate the response by checking: 1. Logical Consistency with the Label
2. Avoidance of Arbitrary Conclusions
3. Internal Coherence
4. Alignment with the Given Evidence

If any of these checks fail, the response should be rated 0.

Document:
{DOCUMENT_PLACEHOLDER}

Statement:
{STATEMENT_PLACEHOLDER}

Model Response:
{MODEL_RESPONSE_PLACEHOLDER}

### Score Rubric (Logical Coherency Evaluation)

Score 0 (Low Logical Coherency): The model's reasoning does not logically support the assigned label. The reasoning process either contradicts the label, includes an arbitrary conclusion, lacks logical progression, or misapplies the classification rules. The model may have arrived at the correct label by luck rather than through valid reasoning, or its explanation contains gaps that prevent a clear justification of the assigned label. If the reasoning indicates one label but the response assigns a different label, or if the reasoning is internally inconsistent, the response must be rated 0.

Score 1 (High Logical Coherency): The model's reasoning process is fully aligned with the assigned label, with each step logically supporting the final classification. The reasoning follows a structured and internally consistent path without contradictions, arbitrary jumps, or gaps. The model applies the labeling criteria correctly, ensuring that the conclusion is reached through a valid and justified reasoning process rather than by chance.

Please review the model response above carefully. Then, output your evaluation in the following format:
Feedback: <detailed feedback explaining your evaluation> [RESULT] <a score between 0 (poor) and 1 (excellent)>

Figure 10: Judge template for Logical Coherency evaluation.

---

**VERIFIABILITY_CHECKER_TEMPLATE**

You are given a single statement. Your job is to classify it as "verifiable," "ambiguous," or "unverifiable" based on the updated criteria below. You must return exactly one of those three words.

Only label a statement verifiable if it is fully self-contained and checkable so that a researcher could confirm or refute the claim using credible sources. If in doubt, use "ambiguous" or "unverifiable," not "verifiable."

### VERIFIABLE

The statement makes a clear, specific factual claim with enough concrete detail to look up or consult sources. The claim does not rely on unclear pronouns ("it," "they," "he"), vague references ("the policy," "this system"), or missing context that prevents checking. The statement is not purely subjective, not a general opinion, and not incomplete/hypothetical.

### AMBIGUOUS

The statement may sound factual, but it is missing crucial details (e.g., which growth factor, which technology, who performed an action, which location). The statement might be time-dependent or context-dependent: we cannot verify it as written without additional context or timeframe. It has some factual or technical flavor, but is too vague to confirm or refute.

### UNVERIFIABLE

The statement is purely opinion, subjective preference, or personal judgment. It is too incomplete or truncated to form a factual claim. It is fully hypothetical with no real anchor. It is excessively broad or unmeasurable.

### EXAMPLES:

VERIFIABLE

"The Eiffel Tower in Paris measures around 324 meters in height."

"All 50 U.S. states require drivers to have a valid license."

"In Homer's Iliad, Achilles slays Hector in Book XXII."

AMBIGUOUS

"This growth factor describes the evolution of large-scale structures in the universe." (Which growth factor? Lacks specificity.)

"In a CRUD system, this is typically done using GET requests or SQL SELECT statements." ("This" is not defined; partial context.)

"It also helps with motor function after 6-OHDA injection in rats." ("It" is undefined; unclear references to a compound or drug.)

UNVERIFIABLE

"Architecture is always marvelous." (Subjective opinion.)

"The new policy, which took effect on..." (Truncated; no testable claim.)

"Vanilla is the best flavor." (Subjective preference.)

### STATEMENT:

{STATEMENT_PLACEHOLDER}

Is this statement verifiable, ambiguous, or unverifiable? Answer with exactly one of the following: "verifiable", "ambiguous", or "unverifiable".

---

Figure 11: Prompt template for statement verifiability evaluation.

| Dataset | Example Unverifiable Sentences |
|---|---|
| RAGTruth | However, it is important to note that this alone is not enough to diagnose a mental illness, and it is not possible to make a definitive diagnosis without more information. |
| | It has a rating of 4.0 stars out of 5.0 based on 3 reviews. |
| TofuEval-MediaS | The band will be inducted anyway. |
| | Employees have 52 days to accept. |
| TofuEval-MeetB | He supported moving cameras from current locations. |
| | It also includes recommendations for future budget planning. |
| Lfqa | This is due to exercise strengthening the heart muscle, enabling it to pump a greater volume of blood with each heartbeat. |
| | The answer is Yes, people do make money from these types of posts. |
| ExpertQA | It can instill incorrect beliefs about resolution of everyday problems through violence and abuse. |
| | It can improve the driver's visibility, safety, and parking ability. |

Table 13: Example unverifiable sentences from source datasets.

---

**Example of Knowledge Ambiguity**

---

**Document:**
The Araona phonemic inventory consists of four vowels (/a/, /e/, /i/, /o/) and 19 consonants.       ···       The consonant table includes nasal, stop, and fricative types. Under palatal nasals, the IPA symbol ɲ is listed, which corresponds to the Spanish letter *ñ* (*eñe*).

**Statement:**
The Araona language uses 19 consonants, including ones of the nasal, stop, and fricative types, as well as the Spanish *ñ*.

**Model Output:**
Reasoning:
1. The document states that the Araona language's phonemic inventory consists of four vowels and 19 consonants. It also provides a table listing its consonants, which are divided into categories such as nasals, stops, and fricatives.
2. The statement mentions that the language uses 19 consonants, including those of nasal, stop, and fricative types. This is supported by the document's explicit description and breakdown of its consonantal inventory.
3. Moreover, the statement specifically refers to the inclusion of the Spanish *ñ*. In the phonemic table, under the nasal category in the palatal column, the consonant [ɲ] appears. [ɲ] is the IPA symbol corresponding to the Spanish letter *ñ*.
Based on these points, the statement "The Araona language uses 19 consonants, including ones of the nasal, stop, and fricative types, as well as the Spanish *ñ*" is strongly supported by the document.
**Final Answer:** [Attributable]

---

Table 14: Qualitative example knowledge ambiguity. The fact verifier model is o3-mini. Model uses external knowledge "[ɲ] is the IPA symbol corresponding to the Spanish letter *ñ*" to verify the statement, which is not specified in the document but is factual.

---

**Example of Linguistic Ambiguity**

---

**Document:**
Irish betting company Paddy Power is backtracking after tweeting that "Newcastle have suffered more Kop beatings over the last 20 years than an unarmed African-American male." The tweet alluded to controversial incidents in the U.S. involving police. A spokesperson said, "It was a joke, and no offense was meant." The tweet was deleted but criticism continued. ⋯ Liverpool beat Newcastle 2-0 in the English Premier League on Monday, with Sterling and Allen scoring the goals.

**Statement:**
Irish betting company Paddy Power apologizes for tweet about Newcastle United. The tweet alluded to recent controversial incidents in the U.S. involving police. The company is well-known for its use of publicity stunts. Liverpool beat Newcastle 2-0 in the English Premier League on Monday.

**Model Output:**
Step-by-step reasoning:
1. The document states that Paddy Power tweeted a controversial message about Newcastle, which alluded to the killing of unarmed African-American men by police in the United States. This directly supports the statement that the tweet referenced recent controversial incidents in the U.S. involving police.
2. The document mentions that the tweet was deleted and that a spokesperson said, "It was a joke, and no offense was meant," indicating that the company backtracked or effectively apologized for the tweet. This supports the claim that Paddy Power apologizes for the tweet.
3. The document also confirms that Paddy Power is well-known for its use of publicity stunts.
4. Additionally, the document reports that Liverpool beat Newcastle 2-0 in the English Premier League on Monday.
Restating the statement: "Irish betting company Paddy Power apologizes for tweet about Newcastle United. The tweet alluded to recent controversial incidents in the U.S. involving police. The company is well-known for its use of publicity stunts. Liverpool beat Newcastle 2-0 in the English Premier League on Monday."
Based on the reasoning above, every part of the statement is supported by the document. [Attributable]

---

Table 15: Qualitative example of linguistic ambiguity. The fact verifier model is o3-mini. "it was a joke, and no offense was meant" could be understood as an apology to someone, or it might not.

---

**Example of Contextual Ambiguity**

**Document:**
[1] Since his 'breakthrough', a famous Nineties photo-shoot for British designer Katharine Hamnett in which a young model sat, legs akimbo, to reveal her pubic hair, he has become the 'go-to guy' for brands seeking to adopt an 'edgy' image. They appear to like him because of his obsession with naturalistic, grungy images of pornified women. "Richardson took 1970s porn aesthetic and made it fashion chic," is how the preface to one of his books describes his career. "Pop stars, supermodels, transsexuals, hillbillies, friends, pets and celebrities all do for his lens what they'll do for no other." ⋯ [2] In the fashion and art worlds—realms that have historically celebrated bohemian, subversive and amoral ideas—reportedly predatory behavior like Richardson's can get covered up by a veneer of glamorous transgression. Artists like Richardson are cast as renegade heroes, capturing something raw and true.

**Statement:**
Terry Richardson is a famous and prolific fashion and portrait photographer, known for his intimate, often provocative shots.

**Model Output:**
Step-by-step reasoning:
1. Summarize the DOCUMENT: - [1] Describes how a famous Nineties photo-shoot by Richardson for designer Katharine Hamnett put him on the map as a "go-to guy" for brands seeking an edgy image. It emphasizes his focus on naturalistic, grungy images and how he transformed a 1970s porn aesthetic into something considered fashion chic. His subjects include pop stars, supermodels, transsexuals, friends, and celebrities.
- [2] Explains that in the realms of fashion and art, often associated with subversive and amoral ideas, photographers like Richardson are seen as renegade heroes. His work is described as raw and true, and his controversial behavior is sometimes overlooked because of his artistic persona.
2. Evaluate the STATEMENT: - The statement says: "Terry Richardson is a famous and prolific fashion and portrait photographer, known for his intimate, often provocative shots." - Evidence from the DOCUMENT indicates that Richardson is indeed famous ("go-to guy" for brands, well-known since his breakthrough photo-shoot) and operates heavily in the fashion realm, capturing edgy and provocative images. The description of his work (e.g., the naturalistic depiction of pornified women and his transformation of aesthetics) directly supports his reputation for provocative, and arguably intimate, portrait-style shots.
3. Conclusion: - The DOCUMENT strongly implies that Richardson is a famous and prolific photographer in fashion and portrait work, known for provocative imagery, which aligns with the STATEMENT provided.
Final Answer: [Attributable]

---

Table 16: Qualitative example of contextual ambiguity. The fact verifier model is o3-mini. The document doesn't specify which "Richardson" it is, while the statement says "Terry Richardson" and the context aligns with the statement.

---

**Example of Numerical Ambiguity**

---

**Document:**
How much does the average American man weigh? The document provides age-based averages:

- 20–39 years: 196.9 pounds
- 40–59 years: 200.9 pounds
- 60 and older: 194.7 pounds

It also states the average American male height is 69.1 inches, and average weight is 197.9 pounds. ⋯ A man who is 69.1 inches tall and weighs 197.9 pounds has a BMI of 29.1 — the high end of the "overweight" classification.

**Statement:**
The average weight of an American male is around 198 pounds.

**Model Output:**
To determine if the STATEMENT is supported by the DOCUMENT, let's break down the relevant information provided:

1. **Average Weight of American Men:** The document provides a breakdown: - 20–39 years: 196.9 pounds - 40–59 years: 200.9 pounds - 60 and older: 194.7 pounds

2. **Overall Average Weight:** Also given as 197.9 pounds.

3. **Comparative Context:** The document discusses increases in average weights over time, and gives a global context, showing North America (including women) at 177.9 pounds.

4. **BMI Insight:** At 197.9 pounds and 69.1 inches tall, BMI is 29.1 — at the high end of "overweight."

5. **Conclusion:** The average weight is stated as 197.9 pounds. While not exactly 198, it is very close and supports the claim that it is "around" 198 pounds.

**Final Evaluation:** The statement "The average weight of an American male is around 198 pounds" is strongly supported by the document.

**Final Answer:** [Attributable]

---

Table 17: Qualitative example of numerical ambiguity. The fact verifier model is GPT-4o. The model correctly calculates the precise number "197.9", but the statement states "around 198" pounds, creating numerical ambiguity.

---

**Example of Missing Fine-grained Detail in Statement**

**Document:**
Colin Kolles (born Călin Colesnic 13 December 1967 in Timișoara, Romania) is the former team principal and managing director of the Hispania Racing F1 Team, previously holding a similar position at the team known under the names Jordan, Midland, Spyker and Force India from 2005 to 2008. He was an advisor to Caterham F1 and had a part in the unsuccessful Forza Rossa Racing project. The Spyker F1 Team, known as the Etihad Aldar Spyker F1 Team for sponsorship reasons, was a Formula One team that competed in the 2007 Formula One World Championship, and was created by Spyker Cars after their buyout of the short-lived Midland F1 (formerly Jordan Grand Prix) team. The change to the Spyker name was accompanied by a switch in racing livery from the red and white previously used by Midland, to an orange and silver scheme—already seen on the Spyker Spyder GT2-R—orange being the national colour and the auto racing colour of the Netherlands. At the end of the 2007 season, the team was sold and renamed Force India. ⋯ In 2014 they lost their Lotus backing and rebranded as ByKolles Racing with a new LMP1 prototype, the CLM P1/01.

**Statement:**
The co-founder of ByKolles Racing and youngest son of Romulus Kolles, is the former team principal and managing director of Spyker F1 that was sold and renamed Force India at the end of 2007.

**Model Output:**
To determine if the STATEMENT is supported by the DOCUMENT, let's break down the information provided in the DOCUMENT and compare it with the STATEMENT:
1. **Colin Kolles**: The DOCUMENT states that Colin Kolles, born Călin Colesnic, is the co-founder of ByKolles Racing, along with his father Romulus Kolles. This aligns with the first part of the STATEMENT regarding Colin Kolles being the co-founder of ByKolles Racing and Romulus Kolles being his father.
2. **Former team principal and managing director roles**: The DOCUMENT mentions that Colin Kolles was the team principal and managing director of the Hispania Racing F1 Team and previously held similar positions at Jordan, Midland, Spyker, and Force India from 2005 to 2008. This part of the DOCUMENT confirms that Colin Kolles was indeed involved as a team principal and managing director for Spyker F1.
3. **Spyker F1 and Force India**: The DOCUMENT explains that Spyker F1 competed in the 2007 Formula One World Championship and was sold and renamed Force India at the end of the 2007 season. This confirms the transition from Spyker F1 to Force India, as mentioned in the STATEMENT.
Now, let's compare this with the STATEMENT: - The STATEMENT claims that the co-founder of ByKolles Racing, who is the youngest son of Romulus Kolles, was the former team principal and managing director of Spyker F1, which was sold and renamed Force India at the end of 2007.
Given the information in the DOCUMENT, all parts of the STATEMENT are supported:
- Colin Kolles is indeed the co-founder and one of the sons of Romulus Kolles. - He was the team principal and managing director of Spyker F1. - Spyker F1 was sold and renamed Force India at the end of 2007.
Therefore, the STATEMENT is strongly supported by the DOCUMENT.
STATEMENT: The co-founder of ByKolles Racing and youngest son of Romulus Kolles, is the former team principal and managing director of Spyker F1 that was sold and renamed Force India at the end of the 2007.
Final Decision: [Attributable]

---

Table 18: Qualitative example illustrating missing fine-grained detail of statement. The fact verifier model used is GPT-4o. Model misses the detail of the statement "youngest son".

| Example of misinterpretation of evidence |
| --- |
| **Document:**
This is the amazing scene as a freelance photographer risks his life to capture an image of a hungry lion from just a few feet away. Fearless Atif Saeed, 38, from Lahore, Pakistan was in the city's safari park when he spotted the male lion in the distance. Mr Saeed got out of his car and crept up towards the hunter armed only with a camera and fired off a few frames. Atif Saeed crept to within 10 feet of the hungry lion before leaving the safety of his car to lie on the ground $\cdots$ the clicking sound of the camera's shutter alerted the lion to Mr Saeed's presence, who was forced to retreat quite rapidly from just 10 feet away from the lion. |
| **Statement:**
Fearless Atif Saeed, 38, from Lahore, Pakistan was in the city's safari park when he spotted the male lion in the distance. Mr Saeed got out of his car and crept up towards the hunter armed only with a camera and fired off a few frames. Unfortunately the clicking sound of the camera's shutter alerted the lion to the hunter's presence. |
| **Model Output:**
Step-by-step reasoning:
1. Main points of the DOCUMENT: - It describes a freelance photographer named Atif Saeed, 38, from Lahore, Pakistan, who was in a city's safari park. - He spotted a male, hungry lion in the distance. - He got out of his car and crept up towards the animal (referred to in the document as "the hunter") armed only with a camera and fired off a few frames. - The clicking sound of the camera's shutter alerted the lion to Atif Saeed's presence, forcing him to retreat from a very close distance.
2. Comparing these points to the STATEMENT: - The STATEMENT mentions that "fearless Atif Saeed, 38, from Lahore, Pakistan was in the city's safari park when he spotted the male lion in the distance," which matches the document. - It also states that "Mr Saeed got out of his car and crept up towards the hunter armed only with a camera and fired off a few frames," aligning well with the document. - Lastly, it states that "unfortunately the clicking sound of the camera's shutter alerted the lion to the hunter's presence." Although the document says the lion was alerted to Mr Saeed's presence, the substitution of "the hunter" still clearly refers to him in the context of the narrative.
Restating the STATEMENT: "Fearless Atif Saeed, 38, from Lahore, Pakistan was in the city's safari park when he spotted the male lion in the distance. Mr Saeed got out of his car and crept up towards the hunter armed only with a camera and fired off a few frames. Unfortunately the clicking sound of the camera's shutter alerted the lion to the hunter's presence."
Based on the reasoning and the supporting evidence from the DOCUMENT, the final answer is:
[Attributable] |

Table 19: Qualitative example illustrating misinterpretation of reference documents. The fact verifier model used is o3-mini. Model misinterprets the meaning of "the hunter" as "Mr. Saeed", where it is clearly indicating "the lion".

---

**Example of Over-generalization**

---

**Document:**
Speaker 0: Report from Public Works recommendation to receive supporting documentation into the record, conclude the public hearing. Find that the area to be vacated is not needed for present or prospective public use and adopt resolution ordering the vacation of the north-south alley west of Long Beach Boulevard between East Waldo Road and 35th Street, and a portion of sidewalk right of way along Locust Avenue, District seven. Speaker 6: Staff have been working closely with Lazar Fish to help them expand their footprint here in the city. They're looking to add more office space.   ⋯
Speaker 2: This alley vacation will allow us to expand and support the incredible work of Lazar Fish, which is an early tech company within the city of Long Beach. They're growing by hundreds of jobs and creating a model facility.      ⋯      Speaker 0: Councilmember Richardson. That's why Richardson motion carries.

**Statement:**
The council unanimously approved vacating an alley to support expansion of Lazar Fish, a local tech company.

**Model Output:**
Step-by-step reasoning:
1. The DOCUMENT is a transcript of a council hearing regarding a resolution to vacate an alley. The key points include: • A recommendation from Public Works to adopt a resolution vacating a north-south alley (and a portion of adjacent sidewalk) along Long Beach Boulevard. • The purpose of the vacating is related to supporting the development efforts of a local company, referred to in the discussion as Lady Fish or Lazar Fish, which is expanding its office space. • Councilmembers express strong support for the project. One councilmember remarked that this move "will allow us to expand and support the incredible work" of Lazar Fish. Others similarly commended the project. • A motion was made, discussed, and ultimately passed ("motion carries"), indicating that the resolution was approved without noted opposition.
2. The STATEMENT states that "The council unanimously approved vacating an alley to support expansion of Lazar Fish, a local tech company." The DOCUMENT supports that: • The council approved a resolution to vacate an alley for the development and expansion of Lazar Fish. • The votes, as indicated by "motion carries" and the lack of any dissenting comments, imply unanimous support.
3. Restating the STATEMENT: "The council unanimously approved vacating an alley to support expansion of Lazar Fish, a local tech company."
4. Given the supportive comments from multiple councilmembers, the passed motion, and the focus on expanding Lazar Fish's operations, the statement is strongly implied by the DOCUMENT.
[Attributable]

---

Table 20: Qualitative example illustrating over-generalization. The fact verifier model used is o3-mini. Model over-generalizes that "motion carreis" indicates "unanimous approval".

---

**Example of a Numerical Error**

---

**Document:**
Volunteer patients were randomized into one of two experiment groups where they would receive an experimental vaccine or a placebo. They were subsequently exposed to a drug-sensitive strain of malaria and observed to see whether they came down with an infection. The vaccine trial data is described in the CSV file malaria.csv.
malaria.csv:
{ "treatment": {0:"vaccine", 1:"vaccine", 2:"vaccine", 3:"vaccine", 4:"vaccine", 5:"vaccine", 6:"vaccine", 7:"vaccine", 8:"vaccine", 9:"vaccine", 10:"vaccine", 11:"vaccine", 12:"vaccine", 13:"vaccine", 14:"placebo", 15:"placebo", 16:"placebo", 17:"placebo", 18:"placebo", 19:"placebo"}, "outcome": {0:"infection", 1:"infection", 2:"infection", 3:"infection", 4:"infection", 5:"no infection", 6:"no infection", 7:"no infection", 8:"no infection", 9:"no infection", 10:"no infection", 11:"no infection", 12:"no infection", 13:"no infection", 14:"infection", 15:"infection", 16:"infection", 17:"infection", 18:"infection", 19:"infection"} }

**Statement:**
The difference in infection rates between the placebo and vaccine groups, rounded to the nearest thousandth, is 0.643.

**Model Output:**
To determine whether the STATEMENT is supported by the DOCUMENT, let's analyze the data provided in the DOCUMENT.
1. **Summarize the Document:** - Volunteers were divided into two groups: one received an experimental vaccine, and the other received a placebo. - After exposure to a drug-sensitive strain of malaria, they were observed for infection outcomes. - The CSV file `malaria.csv` provides specific data on treatments and outcomes.
2. **Extract data from `malaria.csv`:** - Vaccine group: 13 individuals (indices 0-13). - Placebo group: 6 individuals (indices 14-19). - Vaccine group infections: 5 infections, 8 no infections. - Placebo group infections: 6 infections, 0 no infections.
3. **Calculate infection rates:** - Vaccine infection rate: $\frac{5}{13}$ ≈ 0.3846 - Placebo infection rate: $\frac{6}{6}$ = 1.0
4. **Calculate the difference in infection rates:** $1.0 - 0.3846 ≈ 0.6154$
5. **Rounded difference to the nearest thousandth:** 0.615
6. **Comparison with the STATEMENT:** - The STATEMENT claims a difference of 0.643. - Calculated and rounded correctly, the difference is 0.615.
**Final Reasoning:** The STATEMENT claims a specific difference in infection rates between placebo and vaccine groups that does not match the provided data. The correctly calculated difference is 0.615. Thus, the STATEMENT is not supported by the DOCUMENT.
**Final Answer:** [Not Attributable]

---

Table 21: Qualitative example illustrating numerical calculation error. The fact verifier model used is GPT-4o. Model inaccurately calculate the infection rate, miscounting the denominator of the vaccine group.

