# OpenReview forum: "Verifying the Verifiers: Unveiling Pitfalls and Potentials in Fact Verifiers"
_colmweb.org/COLM/2025/Conference — COLM 2025_

### Official Review · Reviewer_TTKa · 2025-04-15

**Rating:** 7
**Confidence:** 4
**Ethics Flag:** 1

**Summary:**

Fact verification is crucial for ensuring reliable LLM applications. In this work, the authors evaluate 13 fact verification models—including frontier and open-weight reasoning LLMs—across 14 fact-checking benchmarks. They report three main findings.

First, they show that about 16% of the data contains annotation errors or ambiguities, which can significantly distort model rankings. To address this, they introduce a scalable LLM-as-a-judge pipeline to detect such issues.

Second, they find that few-shot prompting with frontier LLMs, often overlooked, yields strong performance and should be used as a baseline in future studies.

Lastly, while smaller fine-tuned models are more efficient, they lag in complex reasoning tasks. The authors show that training with synthetic multi-hop data can substantially boost their performance.

**Reasons To Accept:**

1. The paper is clearly written and easy to understand.

2. It effectively points out the limitations of existing fact verification benchmarks.

3. It proposes ClearFacts and GrayFacts, leveraging a well-constructed framework that combines LLM-as-a-judge and human annotator collaboration.

**Reasons To Reject:**

When reviewing the data from fact verification benchmarks, the performance of LLM-as-a-judge was not separately evaluated. Conducting an evaluation of this performance would strengthen the argument for the framework's reliability. If the performance were shown to exceed 95%, it would make the claim that 16% of the dataset is mislabeled significantly more convincing.

---

> ### Author Response · Authors · 2025-05-30
>
> We sincerely appreciate the reviewer’s acknowledgment of the paper’s clear writing, meaningful contributions, and the effectiveness of our analysis.
>
> We have addressed your concerns in the following response. If you still have remaining concerns about the paper, please let us know, and we will do our best to address them.
>
> ### **Evaluation of LLM-as-a-judge framework.**
>
> During our studies, we found that LLM judges often make mistakes. Thus, we didn’t choose to fully automate the pipeline of detecting mislabeled or ambiguous cases from the fact verification benchmarks. Instead, we used LLM judges to curate examples with high potential to be mislabeled or ambiguous, and then manually verified those examples. In other words, the primary purpose of incorporating LLM-as-a-judge in the annotation process was to reduce the human annotation workload (as mentioned in L156 of the manuscript, it took approximately four minutes to annotate each sample, which is very time-consuming to annotate all samples manually).
>
> We agree that evaluating the LLM judge’s performance would improve the reliability of the pipeline, enabling us to identify such cases with less human effort. We appreciate the comment and believe that investigating the performance of LLM judges is a promising direction for future work.

---

> > ### Comment · Reviewer_TTKa · 2025-06-04
> >
> > Thank you for your response. I will keep the score as it is.

---

### Official Review · Reviewer_i78y · 2025-05-12

**Rating:** 6
**Confidence:** 4
**Ethics Flag:** 1

**Summary:**

This paper evaluates 13 fact verification models using data from 14 fact-checking benchmarks and presents three key findings. First, approximately 16% of the dataset labels are either ambiguous or incorrect, which compromises the reliability of model comparisons. To address this, the authors propose a framework that applies a two-stage filtering process combined with LLM-as-a-judge to refine the data. This approach results in two new benchmarks: CLEARFACTS and GRAYFACTS, which are used in subsequent experiments and analysis. Second, advanced commercial LLMs using few-shot prompts outperform many existing models, yet they are often excluded from evaluations. Third, smaller models encounter difficulties with complex reasoning tasks such as multi-hop fact-checking, but their performance improves significantly when synthetic training data is introduced.

**Questions To Authors:**

- For datasets with three labels, the paper maps two labels (not attributable and contradictory) into a single label.
However, in the case of SciFact, which contains a NoInfo label, how is this handled? Are examples with NoInfo excluded?

**Reasons To Accept:**

- This paper provides an in-depth analysis of annotation errors and ambiguity, issues that are often overlooked in previous automated fact-checking research despite their potential to significantly distort results.

- The resulting CLEARFACTS and GRAYFACTS can be valuable to the community.

- The method for generating synthetic multi-hop reasoning data shows promise. Experimental results indicate that this approach is effective for enhancing the performance of a small model.

**Reasons To Reject:**

- The phrasing "13 different fact verification models" may be somewhat misleading.  With the exception of the specialized fact verification model (MiniCheck), the remaining models appear to be pre-trained large language models evaluated in zero-shot and few-shot settings.

- The paper does not differentiate between the judge models and the test models. Four large language models (o3-mini, GPT-4o, Gemini 129 2.0-Flash, and Llama3.1 405B FP8) are used as judges (line 128-129). However, these same models are later evaluated on the refined datasets that they helped curate (Figure 3), which introduces concerns of circularity.

- It is difficult to ensure fairness when comparing a specialized model (such as MiniCheck) with commercial LLMs, as the training data for these commercial models is unknown and may potentially include the correct answers.

- The claim regarding the effectiveness and simplicity of few-shot prompting could be better supported (lines 60–62 and line 232). The few-shot prompts are manually crafted and fixed. The paper does not examine how sensitive the results are to these specific examples. Including an ablation study to evaluate the effects of prompt selection would strengthen the analysis.

---

> ### Author Response · Authors · 2025-05-30
>
> We truly appreciate the reviewer for acknowledging the contribution of our work to the community, as well as the effectiveness of our method of synthetic multi-hop reasoning data. We have addressed your concerns in the following response.
>
> ### **The phrasing of “13 different fact verification models".**
>
> We thank the reviewer for the valuable suggestion. We agree that the phrasing can lead to certain confusion, and commit to revising the term in the camera-ready version to “12 pre-trained LLMs and one specialized fact verifier” for clarity.
>
> ### **Circularity issue of Judges and the tested models.**
>
> We would like to clarify several points to address potential concerns about circularity in our experiments.
>
> First, the four frontier LLMs (``o3-mini``, ``GPT-4o``, ``Gemini 2.0-Flash``, and ``Llama3.1 405B FP8``) we mentioned in L128–L129 (which is mentioned in your response) are not "judge" LLMs, but they are "fact verifier" LLMs. They are prompted in the zero-shot setting (same setting as Table 4) to generate candidate responses (for fact verification), and the responses are evaluated by separate judge LLMs (using ``gpt-4o-mini-2024-07-18`` to prevent potential circularity or judge bias). We select all examples that **any one** of the four fact verifier LLMs got wrong (~40% of the total data), and three judge LLMs evaluate *Completeness*, *Logical Coherence*, and *Faithfulness* of the fact verifier LLM responses, respectively. We have further provided the details of judge implementation in Appendix C.1, and provided the judge templates in Figures 8,9,10 of the Appendix.
>
> Next, we would like to emphasize that to prevent any bias from using the four LLMs’ zero-shot outputs to filter the problematic examples, we specifically excluded these four fact verifier LLMs when evaluating the rankings of different zero-shot LLMs and MiniCheck in Table 4. As for Figure 3, we included these four LLMs in the results because the main purpose of the figure was to compare the effectiveness of few-shot prompting against the zero-shot prompting (i.e., comparison between using the same model).
>
> We will revise the manuscript to make (1) the distinction between judge and fact-verification models and (2) our rationale for including the four models in Figure 3 clearer to readers.
>
> ### **Comparing Minicheck against commercial LLMs.**
>
> We strongly agree that no evaluation can be perfectly fair when we don’t know the actual details of commercial LLMs. However, omitting them prevents us from measuring the true SOTA, and understanding how far smaller and specialized models like MiniCheck have come.
>
> As demonstrated in our paper and previous works, MiniCheck is indeed a very strong model, surpassing the performance of several strong zero-shot LLMs. However, when compared to few-shot prompted LLMs (which were omitted in previous studies), there’s still room for improvement. We believe that reporting these stronger commercial baselines gives the community a realistic target to drive progress toward better specialized models.
>
> ### **Regarding the generalizability of few-shot examples.**
>
> Thank you for the good suggestion! To further demonstrate the generalizability of our few-shot examples, we provide additional experimental results with four additionally crafted few-shot sets, across two proprietary LLMs (``GPT-4o`` and ``Gemini 2.0-Flash``) and two open LLMs (``Llama 3.1-8B Inst``, ``Qwen 2.5-32B Inst``), to ensure the results generalize across different models. Our results show that the LLMs show little variance across performances using different few-shot examples, and some results even outperform the numbers in the paper, indicating that with more careful curation of few-shot examples, practitioners may yield even better results.
>
> | Models | Zero-shot | Few-shot | Run 1 | Run 2 | Run3 | Run4 | Mean | Variance |
> |----------|----------|----------|----------|----------|----------|----------|----------|----------|
> | Gemini 2.0-Flash | 82.0 | 86.0 |85.3|85.3|85.9|85.5 | 85.5 | 0.06|
> | GPT-4o | 83.3 | 87.3 |87.4|87.0|87.0|86.4| 87.0 | 0.128 |
> | Qwen 2.5-32B Inst    | 81.2 | 83.9 |      83.5    |     84.7     |83.7|83.7| 83.9 | 0.22 |
> | Llama 3.1-8B Inst    | 67.2 | 73.9 | 74.8         |  75.1        |  74.3        |    73.4      | 74.4 | 0.415|
>
>
> We hope the results have addressed your concerns, and we will include these results in the updated manuscript. For additional explanation on how the few-shot examples were created, please refer to “Regarding the generalizability of few-shot examples and their crafting details.” section of Reviewer 6Tms’s rebuttal response.
>
> ### **Mapping of “NoInfo” label of Scifact.**
>
> Thanks for the comment. We mapped the “NoInfo” label of SciFact to the “Not Attributable" label. We will revise our paper to include the details about the label mapping.
>
> We thank the reviewer for the constructive suggestions to improve our paper, and we will include the extra experimental results in the Appendix.

---

> > ### Comment · Reviewer_i78y · 2025-06-04
> >
> > Thank you for the detailed clarifications and additional results.
> > I appreciate the explanation regarding the distinction between judge and fact-verification models. It appears I misread the initial setup, particularly regarding the use of separate models for evaluation versus generation.
> >
> > The additional few-shot experiments are helpful in demonstrating robustness across prompt sets. Including these results and some explanation of prompt construction in the revised version will strengthen the paper.
> >
> > Assuming the promised revisions are incorporated to improve clarity on model roles and evaluation design, I have raised my score to 6.

---

### Official Review · Reviewer_NJrb · 2025-05-12

**Rating:** 7
**Confidence:** 3
**Ethics Flag:** 1

**Summary:**

This paper presents a comprehensive reassessment of fact‑verification benchmarks and models. The authors combine 14 publicly‑released datasets (1,749 examples) and uncover that ~16 % of instances suffer from either mis‑labels or genuine ambiguity. They introduce a scalable LLM‑as‑a‑judge pipeline, refine the data into CLEARFACTS (clean) and GRAYFACTS (ambiguous) subsets, and re‑evaluate 13 verifiers ranging from frontier LLMs (GPT‑4o, o1, Claude 3.7, etc.) to the 7 B fine‑tuned MiniCheck. Three key findings emerge: (i) dataset noise can flip model rankings; (ii) few‑shot prompting for frontier LLMs—often omitted—yields state‑of‑the‑art macro‑F1; (iii) small verifiers lag on multi‑hop cases, but synthetic reasoning data narrows the gap. The work concludes with a call for benchmark hygiene and stronger baselines, and releases code, data, and prompts.

**Questions To Authors:**

N/A

**Reasons To Accept:**

* Timely benchmark hygiene. Quantifying and correcting 16 % noisy examples across multiple popular datasets is a valuable community service that will prevent misleading leaderboard conclusions.
* Actionable best‑practice recommendations. Showing that nine hand‑crafted in‑context examples lift every tested LLM supplies an easy‑to‑adopt baseline for future papers.
* Practical recipe for small models. The synthetic multi‑hop data generation method boosts an 8 B verifier by +12.8 F1 on CoverBench and +7.1 F1 on Hover without harming easier sets.

**Reasons To Reject:**

* Limited methodological novelty. The LLM‑judge‑plus‑human‑confirmation pipeline is an incremental combination of existing ideas (LLM‑as‑a‑judge, error spotting, human validation) rather than a fundamentally new algorithm.

---

> ### Author Response · Authors · 2025-05-30
>
> We are grateful to the reviewer for acknowledging the significance of our contributions to the community and the practical value our findings may provide for future practitioners.
>
> We have addressed your concern in the following response. If you still have remaining concerns about the paper, please let us know, and we will do our best to address them.
>
> ### **Limited methodological novelty.**
>
> As you have thankfully pointed out, our main contribution lies in providing a thorough analysis of fact verification benchmarks and models using our crafted CLEARFACTS and GRAYFACTS, thereby providing effective guidelines for future practitioners studying this field. We agree with you that the LLM-judge + human verification pipeline is not a fundamentally new approach, and we also didn’t intend to highlight the novelty of this pipeline in the paper. However, this significantly reduces human efforts by collecting “potential candidates” for benchmark errors, which introduces a meaningful and effective practice.
>
> Also, as Reviewer *6Tms* and *i78y* mentioned, we want to highlight the technical contribution and novelty of generating multi-hop reasoning data, which boosted small model performance by up to **12.8%**. This also provides a promising direction for future research.

---

> > ### Comment · Reviewer_NJrb · 2025-06-06
> >
> > Thank for the reply. That help address my concern and I am willing to increase my score.

---

### Official Review · Reviewer_6Tms · 2025-05-12

**Rating:** 7
**Confidence:** 4
**Ethics Flag:** 1

**Summary:**

The paper collects a diverse benchmark (aggregated from several existing ones) for fact verification and presents key empirical findings through a series of experiments. It evaluates various fact verification models, highlighting the impact of annotation errors, the effectiveness of few-shot prompting, and strategies to improve small language models on this task.

**Questions To Authors:**

- Line 235: “Few-shot prompting has proven to be a simple yet effective technique across many NLP tasks. To explore this, we craft nine in-context examples and use the exact same set across all LLMs evaluated.”
What are these few-shot examples, and how were they crafted? Is it generalizable? This is important information and should be included in the main paper.

- The few-shot prompting experiment is conducted on CLEARFACT. How about GRAYFACT? Do the few-shot gains hold there as well? This is important, especially since your claim suggests that “Few-shot prompted frontier LLMs are strong yet overlooked baselines” — but it's unclear whether this holds for ambiguous cases.

**Reasons To Accept:**

A sound exploratory study: assembling a solid benchmark, proposing a method that carefully distinguishes between Clear and Ambigous (more challenging) cases while highlighting errors in labelling of benchmarks, and experimenting with several models (mainly LLMs) for fact verification, leading to relevant and practical findings:

- Proposes a method to detect labeling issues, enabling both manual correction and identification of ambiguous examples to be treated as more challenging datasets. The approach uses few LLMs as scalable, automated judges to systematically detect these issues and demonstrates how they affect model evaluation.
- Shows that most LLMs, when given just a few in-context examples, perform significantly better, highlighting their capability even with minimal task-specific fine-tuning.
- Shows that integrating synthetic, multi-hop reasoning data into training markedly improves the reasoning ability of smaller LLMs.

The paper is well-written and clearly structured.

**Reasons To Reject:**

Not necessarily a reason to reject, but two of the key findings reported by the authors are validated only on the CLEARFACT dataset. It would strengthen the work to show whether these findings hold on GRAYFACT, which is a contribution of the paper for highlighting ambiguity and challenge in fact verification.

---

> ### Author Response · Authors · 2025-05-30
>
> We sincerely thank the reviewer for acknowledging the soundness and the significance of our study on fact verification benchmarks and models, along with recognizing the effectiveness of our synthetic multi-hop reasoning data. We are also very happy to hear that the paper is well-written.
>
> We have addressed your concerns in the following response. If you still have remaining concerns about the paper, please let us know, and we will do our best to address them.
>
> ### **Generalization of our findings on GRAYFACTS.**
>
> We believe you are questioning the generalization of our findings 2 and 3, on GRAYFACTS.
>
> First, for **Finding 2**: “Few-shot prompting significantly improves the performance of LLM-as-fact-verifiers”, we provide the following table to present that our findings hold.
>
> | Models  | Zero-shot | Few-shot |
> |----------|----------|----------|
> | Llama 3.1-8B Inst |13.5 | 26.5 |
> | Qwen 2.5-32B Inst      | 7.5     | 12.6 |
> | Llama 3.3-70B Inst     | 8.2    | 19.7 |
> | R1-Qwen 2.5-32B Inst | 8.7 | 19.4 |
> | R1-Llama 3.3-70B Inst | 11.1 | 22.9 |
> | Claude 3.5-Haiku | 10.6 | 19.3 |
> | Claude 3.7-Sonnet | 26.7 | 34.3 |
> | Llama 3.1-405B Inst | 10.7 | 27.2 |
> | Gemini 2.0-Flash | 11.3 | 17.7 |
> | GPT-4o | 6.2 | 23.2 |
> | o3-mini | 6.1 | 27.8 |
> | o1 | 9.4 | 25.4 |
>
> The experimental results demonstrate that the use of Few-shot examples **consistently improves** model performance compared to Zero-shot in all models, when tested on GRAYFACTS. We thank the reviewer for the constructive suggestion and will include these results in the next revision. We hope these results address the reviewer’s concern.
>
>
> Next, for **Finding 3**: “A small fine-tuned fact verifier shows limited capabilities on examples requiring complex reasoning”, we show that our synthetic multi-hop data still provides performance gains on GRAYFACTS. We train one 8B Fact Verifier with only ANLI data, and train another with our synthetic multi-hop data and ANLI data, which is the identical setup with Table 5 of the manuscript. The table below shows the results.
>
> | Train Data | Macro F1 |
> |----------|----------|
> | ANLI | 22.4 |
> | ANLI + multi-hop | 24.7 **(+10.3%)**|
>
> The results demonstrate that adding our synthetic multi-hop data improves the performance of the small model by 10.3% in GRAYFACTS, demonstrating that our synthetic data also provides performance gains on the ambiguous dataset.
>
> ### **Regarding the generalizability of few-shot examples and their crafting details.**
>
> Thank you for the good suggestion! We will include the details about how we crafted the few-shot examples, along with the actual few-shot examples. We also commit to including them when we share our codebase.
>
> To craft the few-shot examples, we randomly select examples from the ANLI (stage3) dataset and our synthetic multi-hop dataset (which are completely decontaminated with the test set), with a fair label distribution of three examples per three labels (“Attributable”, “Not Attributable”, “Contradictory”). Next, we use the zero-shot reasoning outputs from models such as ``Llama3.1-405B Instruct FP8``, ``GPT-4o``, and use them as seeds to further verify and refine for actual usage.
>
> To further demonstrate the generalizability of our few-shot examples, we provide additional experimental results with four additionally crafted few-shot sets, across two proprietary models (``GPT-4o`` and ``Gemini 2.0-Flash``) and two open models (``Llama 3.1-8B Inst``, ``Qwen 2.5-32B Inst``), to ensure the results generalize across different models. Our results show that the models show little variance across performances using different few-shot examples, and some results even **outperform** the numbers in the paper, indicating that with more careful curation of few-shot examples, practitioners may yield even better results.
>
> | Models | Zero-shot | Few-shot | Run 1 | Run 2 | Run3 | Run4 | Mean | Variance |
> |----------|----------|----------|----------|----------|----------|----------|----------|----------|
> | Gemini 2.0-Flash | 82.0 | 86.0 |85.3|85.3|85.9|85.5 | 85.5 | 0.06|
> | GPT-4o | 83.3 | 87.3 |87.4|87.0|87.0|86.4| 87.0 | 0.128 |
> | Qwen 2.5-32B Inst    | 81.2 | 83.9 |      83.5    |     84.7     |83.7|83.7| 83.9 | 0.22 |
> | Llama 3.1-8B Inst    | 67.2 | 73.9 | 74.8         |  75.1        |  74.3        |    73.4      | 74.4 | 0.415|
>
>
> We thank the reviewer for the valuable suggestions to improve our paper, and we will also include the extra experimental results in the Appendices.

---

> > ### Comment · Reviewer_6Tms · 2025-06-03
> >
> > Thank you for the additional insights.

---

### Decision · Program_Chairs · 2025-07-08

**Decision:**

Accept

**Comment:**

The paper proposes an assessment of fact verification datasets that is very timely and well-executed. All reviewers agreed to this, and the rebuttals by the authors addressed their concerns.